# MODELS THAT PROVE THEIR OWN CORRECTNESS

## ABSTRACT

How can we trust the correctness of a learned model on a particular input of interest? Model accuracy is typically measured *on average* over a distribution of inputs, giving no guarantee for any fixed input. This paper proposes a theoretically-founded solution to this problem: to train *Self-Proving models* that prove the correctness of their output to a verification algorithm $V$ via an Interactive Proof. We devise a generic method for learning Self-Proving models, and we prove convergence bounds under certain assumptions. Empirically, our learning method is used to train a Self-Proving transformer that computes the Greatest Common Divisor (GCD) *and* proves the correctness of its answer.

## 1 INTRODUCTION

Bob is studying for his algebra exam and stumbles upon a question $Q$ that he cannot solve. He queries a Large Language Model (LLM) for the answer, and it responds with a number: 42. Bob is aware of recent research showing that the LLM attains a 90% score on algebra benchmarks (cf. Frieder et al. 2023), but should he trust that the answer to his particular question $Q$ is indeed 42?

Bob could ask the LLM to explain its answer in natural language. Though he must proceed with caution, as the LLM might try to convince him of an incorrect answer (Turpin et al., 2023). Moreover, even if 42 is the correct answer, the LLM may fail to produce a convincing proof (Wang et al., 2023). If only the LLM could formally prove its answer, Bob would verify the proof and be convinced.

This paper initiates the study of *Self-Proving models* (Fig. 1) that prove the correctness of their answers via an Interactive Proof system (Goldwasser et al., 1985). Self-Proving models successfully convince a verification algorithm $V$ with *worst-case soundness guarantees*: for any question, $V$ rejects all incorrect answers with high probability over the interaction. This guarantee holds even against provers that have access to $V$'s specification, and unbounded computational power.

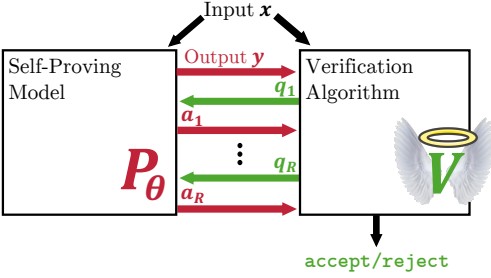

| | Guarantee | Type | Def. |
|---|---|---|---|
| $V$ | Completeness & Soundness | Worst-case $\forall x, y$ | 3.2 |
| $P_\theta$ | Verifiability | Average-case $x \sim \mu$, $y \sim P_\theta(x)$ | 3.4 |

Figure 1: **Self-Proving models.** For input $x$, Self-Proving model $P_\theta$ generates an output $y$ and sends it to a Verification Algorithm $V$. Then, over $i \in [R]$ rounds, $V$ sends query $q_i$, and receives an answer $a_i$ from $P_\theta$. Finally, $V$ decides ("accept/reject") whether it is convinced that $y$ is a correct output for $x$.

Table 1: **Formal guarantees.** Completeness and soundness are fundamental guarantees of a verification algorithm $V$. Verifiability (novel in this work) is a feature of a model $P_\theta$ with respect to a verifier $V$ and input distribution $\mu$. Importantly, $V$'s soundness holds for any input $x$ and output $y$.

| Learning method | Correctness (%) | Verifiability (%) |
|---|---|---|
| GPT (baseline) | 99.8 | - |
| GPT+TL | 98.8 | 60.3 |
| GPT+TL+RLVF | 98.9 | 78.3 |
| GPT+Annotated TL | 98.6 | 96.0 |

Table 2: **Self-Proving transformers computing the GCD.** We train a 6.3M parameter GPT to compute the GCD of two integers sampled log-uniformly from $[10^4]$. Vanilla GPT correctly generates the GCD for almost all inputs, but does not prove correctness to a simple verification algorithm. GPT trained with Transcript Learning (+TL) proves its answer 60.3% of the time; adding Reinforcement Learning from Verifier Feedback (+RLVF) increases this to 78.3%; instead training with Annotated Transcript Learning gives the highest Verifiability score of 96%. See Section 5 for details.

Our contributions are as follows.

- We define Self-Proving models (Section 3).

- We propose two methods for learning Self-Proving models in Section 4. The first, *Transcript Learning (TL)*, relies on access to transcripts of accepting interactions and is the focus of this paper; we prove convergence bounds for TL under convexity and Lipschitzness assumptions. The second method, *Reinforcement Learning from Verifier Feedback (RLVF)*, trains a model by emulating interaction with the verifier. We also present variants of these algorithms that use *Annotations* to improve learning in practice.

- We empirically study TL and Annotated-TL (ATL) for training Self-Proving transformers that compute the Greatest Common Divisor (GCD) of two integers. Table 2 demonstrates the efficacy of our methods, with additional experiments in Section 5. Our results may be of independent interest for research on the arithmetic capabilities of transformers (e.g. Charton 2024; Lee et al. 2024). Code, data and models are available as supplementary material.

**Scope.** This paper contains a theory of learned models that prove their own correctness via an Interactive Proof system. The fascinating and well-studied question of *which* settings are verifiable in an Interactive Proof system is beyond our scope. Our theory is general in that it pertains to *any* such setting, e.g., any decision problem solvable in polynomial space (Shamir, 1992). See Goldreich (2008) for a primer on Proof systems more broadly.

## 2 RELATED WORK

This paper is situated at the intersection of machine learning (ML) and Interactive Proof systems (IPs). We briefly discuss recent relevant work from these literatures.

**ML and IPs.** IPs have found numerous applications in ML towards a diverse set of goals. Anil et al. (2021) introduce Prover–Verifier Games (PVGs), a game-theoretic framework for learned provers and learned verifiers. PVGs were further investigated in at least two subsequent works: Hammond & Adam-Day (2024) study multi-prover and Zero Knowledge variants of PVGs. Additionally, Kirchner et al. (2024) successfully utilize PVGs towards obtaining human-legible outputs from LLMs. Notably, they require a relaxed completeness guarantee of their learned proof system—this requirement is the same as our Definition 3.4 of Self-Proving models.

Beyond PVGs, Wäldchen et al. (2024) cast the problem of model interpretability as a Prover–Verifier interaction between a learned feature selector and a learned feature classifier. Debate systems (Condon et al., 1995), a multiprover variant of IPs, were considered for aligning models with human values (Irving et al., 2018; Brown-Cohen et al., 2023). In such Debate systems, two competing models are each given an alleged answer $y \neq y'$, and attempt to prove the correctness of their answer to a (human or learned) judge. Lastly, Murty et al. (2023) define Pseudointelligence: a model learner $L_M$ and an evaluator learner $L_E$ are each given samples from a ground-truth; $L_M$ learns a model of the ground-truth, while $L_E$ learns an evaluator of such models; the learned evaluator then attempts to distinguish between the learned model and the ground-truth in a Turing Test-like interaction.

All of these works consider *learned verifiers*, whereas our work focuses on training models that interact with a manually-defined verifier. More related in this regard is IP-PAC (Goldwasser et al., 2021), in which a learner proves that she learned a model that is Probably Approximately Correct (Valiant, 1984). We, however, consider *models* that prove their own correctness on a *per-input basis*, rather than *learners* that prove *average-case correctness* of a model.

**Models that generate formal proofs.** Self-Proving models are verified by an algorithm with formal completeness and soundness guarantees (see Definition 3.2). In this sense, Self-Proving models generate a formal proof of the correctness of their output. Several works propose specialized models that generate formal proofs.

AlphaGeometry (Trinh et al., 2024) is capable of formally proving olympiad-level geometry problems; Others have trained models to produce proofs in Gransden et al. (2015); Polu & Sutskever (2020) and others train models to produce proofs in Coq (Gransden et al., 2015), Metamath (Polu & Sutskever, 2020), Lean (Yang et al., 2023), or manually-defined deduction rules (Tafjord et al., 2020); FunSearch (Romera-Paredes et al., 2024) evolves LLM-generated programs by systematically evaluating their correctness. Indeed, all of these can be cast as Self-Proving models developed for *specific proof systems*. Meanwhile, this work defines and studies the class of such models *in general*. Several works (e.g. Welleck et al. 2022) consider models that generate natural language proofs or explanations, which are fundamentally different from formal proofs (or provers) verified by an algorithm.

**Training on intermediate steps.** Chain-of-Though (CoT, Wei et al. 2022) refers to additional supervision on a model in the form of intermediate reasoning steps. CoT is known to improve model performance whether included in-context (Wei et al., 2022) or in the training phase itself (Yang et al., 2022). Transcript Learning (TL, Section 4.1) can be viewed as training the model on a Chain-of-Thought induced by the interaction of a verifier and an honest prover (Definition 3.2).

To complete the analogy, let us adopt the terminology of Uesato et al. (2022), who consider *outcome supervision* and *process supervision*. In our case, the *outcome* is the decision of the verifier, and the *process* is the interaction between the verifier and the model. Thus, Reinforcement Learning from Verifier Feedback (RLVF, Section 4.2) is outcome-supervised while TL is process-supervised. In a recent work, Lightman et al. (2024) find that process-supervised transformers outperform outcome-supervised ones on the MATH dataset (Hendrycks et al., 2021).

**Transformers for arithmetic.** In Section 5 we train and evaluate Self-Proving transformers to generate the GCD of two integers and prove its correctness to a verifier. These experiments leverage a long line of work on neural models for arithmetic tasks originating with Siu & Roychowdhury (1992), and in particular modular arithmetic, which is known to be challenging (Palamas, 2017). Of particular relevance is the recent paper of Charton (2024), who trains transformers to generate the GCD—without a proof of correctness. We benefit from conclusions suggested in their work and start from a similar (scaled-down) experimental setup. Our main challenge (obtaining *Self-Proving* models) is overcome by introducing Annotated Transcript Learning (ATL).

We conduct ablation experiments to find two deciding factors in ATL. First, we study the effect of the amount of annotation given in the form of intermediate steps (Lee et al., 2024), which is related to autoregressive length complexity (Malach, 2023). Second, we characterize ATL efficacy in terms of an algebraic property of the tokenization scheme (cf. Nogueira et al. 2021; Charton 2022; 2024).

## 3 SELF-PROVING MODELS

We introduce and formally define our learning framework in which models prove the correctness of their output. We start with preliminaries from the learning theory and proof systems literatures in Section 3.1. We then introduce our main definition in Section 3.2.

### 3.1 PRELIMINARIES

Let $\Sigma$ be a finite set of tokens and $\Sigma^*$ denote the set of finite sequences of such tokens. We consider sequence-to-sequence models $F_\theta : \Sigma^* \to \Sigma^*$, which are total functions that produce an output for each possible input sequence. A model is parameterized by a real-valued, finite dimensional vector $\theta$.

We consider models as *randomized* functions, meaning that $F_\theta(x)$ is a random variable over $\Sigma^*$, of which samples are denoted by $y \sim F_\theta(x)$.

Before we can define models that prove their own correctness, we must first define correctness. Correctness is defined with respect to an input distribution $\mu$ over $\Sigma^*$, and a ground-truth $F^*$ that defines correct answers. For simplicity of presentation, we focus on the case that each input $x \in \Sigma^*$ has exactly one correct output $F^*(x) \in \Sigma^*$, and a zero-one loss function on outputs (the general case is deferred to Appendix A). The fundamental goal of machine learning can be thought of as learning a model of the ground-truth $F^*$. Formally,

**Definition 3.1** (Correctness). *Let $\mu$ be a distribution of input sequences in $\Sigma^*$ and let $F^*: \Sigma^* \to \Sigma^*$ be a fixed (deterministic) ground-truth function. For any $\alpha \in [0,1]$, we say that model $F_\theta$ is $\alpha$-correct (with respect to $\mu$) if*

$$\Pr_{\substack{x \sim \mu \\ y \sim F_\theta(x)}} [y = F^*(x)] \geq \alpha.$$

An *interactive proof system* (Goldwasser et al., 1985) is a protocol carried out between an efficient *verifier* and a computationally unbounded *prover*. The prover attempts to convince the verifier of the correctness of some assertion, while the verifier accepts only correct claims. The prover is powerful yet untrusted; in spite of this, the verifier must reject false claims with high probability.

In the context of this work, it is important to note that the verifier is *manually-defined* (as opposed to learned). Formally, the verifier is a probabilistic polynomial-time algorithm tailored to a particular ground-truth capability $F^*$. Informally, the verifier is the anchor of trust: think of the verifier as an efficient and simple algorithm, hosted in a trustworthy environment.

Given an input $x \in \Sigma^*$, the model $F_\theta$ "claims" that $y \sim F_\theta(x)$ is correct. We now define what it means to *prove* this claim. We will use $P_\theta$ to denote Self-Proving models, noting that they are formally the same object[1] as non-Self-Proving ("vanilla") models $F_\theta$. This notational change is to emphasize that $P_\theta$ first outputs $y \sim P_\theta(x)$ *and is then prompted by the verifier*, unlike $F_\theta$ who only generates an output $y \sim F_\theta(x)$.

A Self-Proving model proves that $y \sim P_\theta(x)$ is correct to a verifier $V$ over the course of $R$ rounds of interaction (Figure 1). In each round $i \in [R]$, verifier $V$ queries $P_\theta$ on a sequence $q_i \in \Sigma^*$ to obtain an answer $a_i \in \Sigma^*$; once the interaction is over, $V$ accepts or rejects. For fixed $x, y \in \Sigma^*$, the decision of $V$ after interacting with $P_\theta$ is a random variable over $V$'s decision (accept/reject), determined by the randomness of $V$ and $P_\theta$. The decision random variable is denoted by $\langle V, P_\theta \rangle (x, y)$.

We present a definition of Interactive Proofs restricted to our setting.

**Definition 3.2.** *Fix a soundness error $s \in (0,1)$, a finite set of tokens $\Sigma$ and a ground-truth $F^*: \Sigma^* \to \Sigma^*$. A verifier $V$ (in an Interactive Proof) for $F^*$ is a probabilistic polynomial-time algorithm that is given explicit inputs $x, y \in \Sigma^*$ and black-box (oracle) query access to a prover $P$.[2] It interacts with $P$ over $R$ rounds (see Figure 1) and outputs a decision $\langle V, P \rangle (x, y) \in \{\text{reject}, \text{accept}\}$. Verifier $V$ satisfies the following two guarantees:*

- Completeness: *There exists an* honest prover $P^*$ *such that, for all $x \in \Sigma^*$,*

$$\Pr[\langle V, P^* \rangle (x, F^*(x)) \text{ accepts}] = 1,$$

  *where the probability is over the randomness of $V$.[3]*

- Soundness: *For all $P$ and for all $x, y \in \Sigma^*$, if $y \neq F^*(x)$ then*

$$\Pr[\langle V, P \rangle (x, y) \text{ accepts}] \leq s,$$

  *where the probability is over the randomness of $V$ and $P$, and $s$ is the soundness error.*

The efficiency of an interactive proof is usually measured with respect to four parameters: the round complexity $R$, the communication complexity (the overall number of bits transferred during

---

[1] Both are randomized mappings from $\Sigma^*$ to $\Sigma^*$.

[2] We intentionally write $P$ rather than $P_\theta$: Interactive Proofs are defined with respect to all possible provers, not just parameterized ones.

[3] WLOG, the honest prover is deterministic by fixing the optimal randomness of a randomized prover.

the interaction), $P^*$'s efficiency and $V$'s efficiency. These complexity measures scale with the computational complexity of computing the ground-truth $F^*$. For example, an interactive proof for a complex $F^*$ may require multiple rounds of interaction.

**Remark 3.3** (Verifier efficiency). *Definition 3.2 requires that $V$ is a polynomial-time algorithm whereas provers are unbounded. This captures a requirement for* efficient verification*. We chose polynomial time as a measure of efficiency because it is common Proof systems literature. That said, one could adapt Definition 3.2 to fit alternative efficiency measures, such as space complexity (Condon & Lipton, 1989) or circuit depth (Goldwasser et al., 2007). Regardless of which measure is taken, to avoid a trivial definition it is crucial that $V$ should be more efficient than the honest prover $P^*$; else, $V$ can simply execute $P^*$ to perform the computation itself.*

By definition, the soundness error $s$ of a verifier $V$ bounds the probability that it is mistakenly convinced of an incorrect output; in that sense, the smaller $s$, the "better" the verifier $V$. In our setting, we think of a manually-defined verifier $V$ who is formally proven (by a human) to have a small soundness error by analysis of $V$'s specification.

As depicted in Figure 1, each of the model's answers depends on all previous queries and answers in the interaction. This captures the setting of *stateful models*, e.g. a session with a chatbot.

Towards defining Self-Proving models (Section 3.2), let us observe the following. Completeness and soundness are *worst-case guarantees*, meaning that they hold for all possible inputs $x \in \Sigma^*$. In particular, completeness implies that for all $x \in \Sigma^*$, the honest prover $P^*$ convinces $V$ of the correctness of $F^*(x)$; in classical proof systems there is no guarantee that an "almost honest" prover can convince the verifier (cf. Paradise 2021). Yet, if we are to *learn* a prover $P_\theta$, we cannot expect it to agree with $P^*$ perfectly, nor can we expect it to always output $F^*(x)$. Indeed, Self-Proving models will have a *distributional guarantee* with respect to inputs $x \sim \mu$.

## 3.2 Self-Proving models

We define the *Verifiability* of a model $P_\theta$ with respect to an input distribution $\mu$ and a verifier $V$. Intuitively, Verifiability captures the ability of the model to prove the correctness of its answer $y \sim P_\theta(x)$, when the input $x$ is sampled from $\mu$. We refer to models capable of proving their own correctness as *Self-Proving models*. Notice that, as in Definition 3.2, the verifier is fixed and agnostic to the choice of the Self-Proving model.

**Definition 3.4** (Self-Proving model). *Fix a verifier $V$ for a ground-truth $F^*: \Sigma^* \to \Sigma^*$ as in Definition 3.2, and a distribution $\mu$ over inputs $\Sigma^*$. The* Verifiability *of a model $P_\theta: \Sigma^* \to \Sigma^*$ is defined as*

$$\text{ver}_{V,\mu}(\theta) \coloneqq \Pr_{\substack{x \sim \mu \\ y \sim P_\theta(x)}} [\langle V, P_\theta \rangle (x, y) \text{ accepts}] . \tag{1}$$

*We say that model $P_\theta$ is $\beta$-Self-Proving with respect to $V$ and $\mu$ if $\text{ver}_{V,\mu}(\theta) \geq \beta$.*

**Remark 3.5** (Verifiability $\implies$ correctness). *Notice that the ground-truth $F^*$ does not appear in Definition 3.4 except for the first sentence. Indeed, once it is established that $V$ is a verifier for $F^*$ (as per Definition 3.2), then Verifiability w.r.t $V$ implies correctness w.r.t $F^*$: Consider any input distribution $\mu$, ground-truth $F^*$, and a verifier $V$ for $F^*$ with soundness error $s$. By a union bound, if model $P_\theta$ is $\beta$-Verifiable, then it is $(\beta - s)$-correct. That is to say, Verifiability is formally a stronger guarantee than correctness when $V$ has small soundness error $s$.*

As depicted in Figure 1, a Self-Proving model $P_\theta$ plays a dual role: first, it generates an output $y \sim P_\theta(x)$, and then it proves the correctness of this output to $V$. Note also that Self-Provability is a feature of a *model*, unlike completeness and soundness which are features of a *verifier* (see Table 1).

The benefit of Verifiability over correctness is captured by the following scenario. Alice wishes to use a model $P_\theta$ to compute some functionality $F^*$ on an input $x_0$ in a high risk setting. Alice generates $y_0 \sim P_\theta(x_0)$. Should Alice trust that $y_0$ is correct? If Alice has a held-out set of labeled samples, she can estimate $P_\theta$'s average correctness on $\mu$. Unfortunately, (average) correctness provides no guarantee regarding the correctness of the particular $(x_0, y_0)$ that Alice has in hand. If, however, Alice has access to a verifier $V$ for which $P_\theta$ is Self-Proving, then she can trust the model on an input-by-input (rather than average-case) basis: Alice can execute $V$ on $(x_0, y_0)$ and black-box access to $P_\theta$. Soundness of $V$ guarantees that if $y_0$ is incorrect, then $V$ rejects with high probability, in which case Alice should either generate $P_\theta(x_0)$ again—or find a better model.

## 4 LEARNING SELF-PROVING AUTOREGRESSIVE MODELS

With a sound verifier $V$ at hand, obtaining Self-Proving models with respect to $V$ holds great promise: a user that prompts the model with input $x$ does not need to take it on good faith that $P_\theta(x)$ is correct; she may simply verify this herself by executing the verification protocol. How, then, can we learn models that are not just approximately-correct, but Self-Proving as well?

The challenge is to align the model with a verifier. We assume that the learner has access to input samples $x \sim \mu$ and correct outputs $F^*(x)$, as well as the verifier specification (code). Additionally, the learner can emulate the verifier, as the latter is computationally efficient (Remark 3.3).

Our focus is on autoregressive sequence-to-sequence (Self-Proving) models $P_\theta$. Such models generate their output by recursively prompting a randomized sampling from a base distribution $p_\theta$ over tokens $\Sigma$. For an input $z \in \Sigma^*$, the output $w \sim P_\theta(z)$ is generated as follows:

- Sample $w_1 \sim p_\theta(z)$.
- Let $j = 1$. While $w_j$ is not the end-of-sequence token $\text{EOS} \in \Sigma$:
    - Sample $w_{j+1} \sim p_\theta(zw_1 \cdots w_j)$.
    - Update $j := j + 1$.
- Output $w = w_1 w_2 \cdots w_j$.

For any $z \in \Sigma^*$, it is useful to consider the vector of log-probabilities over $\Sigma$, denoted by $\log p_\theta(z) \in \mathbb{R}^{|\Sigma|}$. We assume that each coordinate in this vector is differentiable with respect to $\theta$.

Our general approach is inspired by Reinforcement Learning from Human Feedback (Christiano et al., 2017), a method for aligning models with human preferences, which has recently been used to align sequence-to-sequence models (Ouyang et al., 2022). However, there are two important differences between humans and algorithmic verifiers: (1) Verifiers are efficient algorithms which may be emulated by the learner. This is unlike humans, whose preferences are costly to obtain. On the other hand, (2) verifiers make a single-bit decision at the end of an interaction, but cannot guide the prover (model) in intermediate rounds. In RL terms, this is known as the *exploration problem* for sparse reward signals (e.g. Ladosz et al. 2022).

Section 4.1 introduces *Transcript Learning* (TL), a learning algorithm that overcomes the exploration problem mentioned in the second point under the assumption that the learner has access to transcripts of interactions in which the verifier accepts. We prove convergence bounds for TL (Appendix B.1) and analyze it experimentally (Section 5).

Access to accepting transcripts is a reasonable assumption, for example, when there is an efficient honest prover that can generate such transcripts (Goldwasser et al., 2015). When there is no access to accepting transcripts, we propose *Reinforcement Learning from Verifier Feedback* (Section 4.2).

### 4.1 TRANSCRIPT LEARNING

We present an algorithm for learning Self-Proving models which uses access to a distribution of accepting transcripts. This is a reasonable assumption to make when the honest prover $P^*$ (see Definition 3.2) is efficient, as in the case of public-coin Doubly-Efficient Interactive Proof systems as defined by Goldwasser et al. (2015) and developed in other theoretical (e.g. Goldreich & Rothblum 2018) and applied (e.g. Zhang et al. 2021) works. In this case, an honest prover $P^*$ can be run by the learner during training to collect accepting transcripts without incurring heavy computational cost. Alternatively, the learner may collect a dataset of accepting transcripts prior to learning (see Figure 4 in Appendix B).

The intuition behind Transcript Learning is that the interaction of the verifier and prover can be viewed as a sequence itself, which is called the *transcript* $\pi \in \Sigma^*$. The idea is to learn a model not just of $x \mapsto y^*$ for a correct output $y^*$, but of $x \mapsto y^*\pi^*$, where $\pi^*$ is a transcript of an interaction in which the verifier accepted.

In more detail, Transcript Learning (TL, Algorithm 1) requires access to an *(honest) transcript generator* $\mathcal{T}^*$. Given an input $x$, the generator $\mathcal{T}^*(x)$ samples a sequence $P^*(x)\pi^* \in \Sigma^*$ such that $\pi^*$ is an accepted transcript. TL trains a Self-Provable model by autoregressively optimizing towards

generating accepting transcripts. At a very high level, it works by repeatedly sampling $x \sim \mu$ and transcript $y^* \pi^* \sim \mathcal{T}^*(x)$, and updating the logits $\log p_\theta$ towards agreeing with $y^* \pi^*$ via Gradient Ascent. We prove that, under certain conditions, it is expected to output a Self-Provable model.

**Theorem 4.1** (Theorem B.5, informal). *Fix an input distribution $\mu$, a verifier $V$, a transcript generator $\mathcal{T}^*$, an autoregressive model family $\{P_\theta\}_\theta$ parameterized by $\theta \in \mathbb{R}^d$ for some $d \in \mathbb{N}$, and a norm $||\cdot||$ on $\mathbb{R}^d$. Assume that the* agreement function $A\colon \mathbb{R}^d \to [0,1]$ *defined by*

$$A(\theta) := \Pr_{\substack{x \sim \mu \\ \pi^* \sim \mathcal{T}^*(x)}} [\mathrm{Transcript}(\langle V, P_\theta \rangle (x)) = \pi^*]$$

*is concave in $\theta$. For any $\varepsilon > 0$, let $B_{\mathrm{Norm}}$, $B_{\mathrm{Lip}}$ and $C$ be upper-bounds such that the following conditions hold.*

- *There exists $\theta^* \in \mathbb{R}^d$ with $||\theta^*|| < B_{\mathrm{Norm}}$ such that $A(\theta^*) \geq 1 - \varepsilon/2$.*

- *For all $\theta$, the logits of $P_\theta$ are $B_{\mathrm{Lip}}$-Lipschitz in $\theta$.*

- *The number of tokens sent by the prover to the verifier $V$ in any interaction is at most $C$.*

*Denote by $\bar{\theta}$ the output of Transcript Learning (Algorithm 1) running for $N$ interations, where*

$$N \geq 4 \cdot C^2 \cdot \frac{B_{\mathrm{Norm}}^2 \cdot B_{\mathrm{Lip}}^2}{\varepsilon^2} \tag{2}$$

*and learning rate $\lambda = B_{\mathrm{Norm}}/CB_{\mathrm{Lip}}\sqrt{N}$. Then the expected Verifiability of $\bar{\theta}$ is at least $1 - \varepsilon$.*

The proof (Appendix B) goes by reduction to Stochastic Gradient Descent (SGD). We show (Lemma B.4) that the learner can use its only available tools—sampling honest transcripts, emulating the verifier, and differentiating the logits—to optimize the agreement $A(\theta)$. Specifically, this is done by accumulating gradients from the cross-entropy loss computed at each token. Since $A(\theta)$ lower bounds the Verifiability of $P_\theta$, the former can be used as a surrogate for the latter.

The conditions for Theorem 4.1 can be split into two. First, the standard conditions used to prove SGD convergence: convexity,[4] $B_{\mathrm{Norm}}$-boundedness, and $B_{\mathrm{Lip}}$-Lipschitzness. Second, there is a bound $C$ on the *communication complexity* of the prover in the Interactive Proof system.

Quantitatively, the efficiency of TL is captured by the *number of iterations $N$*. It is desirable to minimize $N$, which is also the *number of samples* needed from the distribution $\mu$ and the transcript generator $\mathcal{T}^*$. The bound on $N$ in Equation (2) can be decomposed into the complexity of SGD ($B_{\mathrm{Norm}}^2 B_{\mathrm{Lip}}^2/\varepsilon^2$), and communication complexity of the proof system $O(C^2)$. Minimizing communication complexity has been an overarching goal in the study of proof systems (e.g. Goldreich & Håstad 1998; Goldreich et al. 2002; Reingold et al. 2021). Theorem 4.1 formally shows the benefit of communication-efficient proof systems in the context of Self-Proving models.

## 4.2 REINFORCEMENT LEARNING FROM VERIFIER FEEDBACK (RLVF)

As mentioned in Section 4.1, Transcript Learning uses access to an honest transcript generator to estimate gradients of (a lower bound on) the Verifiability of a model $P_\theta$.

*Reinforcement Learning from Verifier Feedback (RLVF, Algorithm 2)* estimates this gradient without access to a transcript generator. RLVF can be viewed as a modification of TL in which the learner emulates the interaction of the verifier with its own model $P_\theta$. Rather than directly sampling from the generator as in TL, it collects accepting transcripts by rejection sampling on emulated transcripts.

This rejection sampling means that RLVF requires its initial model $P_{\theta_0}$ to have Verifiability bounded away from 0, so that accepting transcripts are sampled with sufficient probability. Fortunately, such a Self-Proving base model can be learned using TL. This gives a learning paradigm in which a somewhat-Self-Proving base model is first learned with TL (with Verifiability $\delta > 0$), and then "amplified" to a fully Self-Proving model using RLVF (cf. Nair et al. 2018).

---

[4]Convexity does not hold in general LLM training. Yet, Theorem 4.1 provides useful theoretical analysis in a simplified setting, which we empirically validate in the non-convex setting in Section 5.

We prove that RLVF learner can estimate the Verifiability gradient of $P_\theta$ using emulation alone in Lemma B.7. From a broader perspective, RLVF can be derived by viewing Self-Proving as a reinforcement learning problem in which the agent (prover) is rewarded when the verifier accepts. Indeed, RLVF is the Policy Gradient method (Sutton et al., 1999) for a verifier-induced reward. Convergence bounds for Policy Gradient methods are a challenging and active area of research (e.g. Agarwal et al. 2021), and so we leave the full analysis to future work.

### 4.3 Learning from annotated transcripts

To minimize the length of messages exchanged in an Interactive Proof system, the honest prover is designed to send the shortest possible message to the verifier, containing only essential information.

However, when training Self-Proving model, it may be useful for it to first generate an "annotated" answer $\widetilde{a}$ which is then trimmed down to the actual answer $a$ to be sent to the verifier. We adapt Sections 3 and 4 to this setting in Appendix D, where we present *Annotated Transcripts*. The TL and RLVF algorithms naturally extend to annotated transcripts as well. Table 2 shows that annotations significantly improve performance of TL.

Annotations can be viewed as adding Chain-of-Thought (Wei et al., 2022). As a concrete example, consider our experiments on computing the GCD. As detailed in Section 5.2, a proof $\pi$ in this setting is the output of an iterative process—the extended Euclidean algorithm—starting from the input $x$: $x \mapsto \pi_1 \mapsto \pi_2 \mapsto \cdots \mapsto \pi$. The annotation of the proof $\pi$ consists the first $T$ steps $(\pi_1, \ldots, \pi_T)$ up to some fixed cutoff $T$. These are prepended to the proof and shown to the model during TL training. At inference time, the model is evaluated only on whether it generated the proof $\pi$ correctly.

## 5 Experimental Results

We describe our experimental setup, and present ablation studies that shed additional light on the effect of *annotation* and *representation* on Verifiability.

### 5.1 Setup: Training transformers to predict the GCD of two integers

Charton (2024) empirically studies the power and limitations of learning GCDs with transformers. We follow their setup and two conclusions on settings that make for faster learning: Training from the log-uniform distribution, and choosing a base of representation with many prime factors.

We fix a base of representation $B = 210$ and use $\mathbf{x}$ to denote an integer $x$ encoded as a $B$-ary string.[5] For sequences of integers, we write $(\mathbf{x_1 x_2})$ to denote the concatenation of $\mathbf{x_1}$ with $\mathbf{x_2}$, delimited by a special token. The vocabulary size needed for this representation is $|\Sigma| \approx 210$.

We choose the input distribution $\mu$ to be the log-uniform distribution on $[10^4]$, and train the transformer on sequences of the form $(\mathbf{x_1 x_2 y})$, where $x_1, x_2 \sim \mu$ and $y = GCD(x_1, x_2)$. This is a scaling-down of Charton (2024), to allow single GPU training of Self-Proving transformers. In all of our experiments, we use a GPT model (Vaswani et al., 2017) with 6.3M parameters trained on a dataset of 1024K samples in batches of 1024. Full details are deferred to Appendix F.

**Proving correctness of GCD.** Following Charton (2024) as a baseline, we find that transformers can correctly compute the GCD with over $99\%$ probability over $(x_1, x_2) \sim \mu$. To what extent can they *prove* their answer? To answer this question, we first devise a natural proof system based on Bézout's theorem. Its specification and formal guarantees are deferred to Appendix E. We denote its verification algorithm by $V$, and highlight some important features of the experimental setup:

- The proof system consists of one round ($R = 1$). The verifier makes no query, and simply receives a proof $\pi$ from the prover.
- *Completeness:* For any $x_1, x_2, y \in [10^4]$ such that $y = GCD(x_1, x_2)$, there exists a proof $\pi$ such that $V(\mathbf{x_1 x_2 y}\pi)$ accepts. As detailed in Appendix E, the proof $\pi$ consists of a pair of integers who are *Bézout coefficients* for $x_1, x_2$.

---

[5]$B = 210$ is chosen following Charton (2024) to be an integer with many prime factors.

- *Soundness:* If $y \neq GCD(x_1, x_2)$, then $V(\mathbf{x_1 x_2 y}\pi)$ rejects[6] for any alleged proof $\pi \in \Sigma^*$.

To measure Verifiability, we train a Self-Proving transformer using Transcript Learning on sequences $(\mathbf{x_1 x_2 y}\pi)$ and estimate for how many inputs $x_1, x_2 \sim \mu$ does the model generate *both* the correct GCD $\mathbf{y}$ and a valid proof $\pi$. We test on 1000 pairs of integers $x_1', x_2' \sim \mu$ held-out of the training set, prompting the model with $(\mathbf{x_1' x_2'})$ to obtain $(\mathbf{y'}\pi')$, and testing whether $V(\mathbf{x_1' x_2' y'}\pi')$ accepts.

Table 2 shows our main experimental result, which has the following key takeaways:

1. Transcript Learning (TL) for 100K iterations ($\approx$100M samples) results in a Self-Proving transformer that correctly proves 60.3% of its answers.

2. A base Self-Proving Model with fairly low Verifiability of 40% can be improved to 79.3% via Reinforcement Learning from Verifier Feedback (RLVF). Although it does not rely on honest transcripts, RLVF trains slowly: this nearly-twofold improvement took four million iterations.

3. Most efficient is Annotated Transcript Learning, with 96% Verifiability in 100K iterations.

We further investigate the effect of annotations next.

## 5.2 MODELS GENERALIZE BEYOND ANNOTATIONS

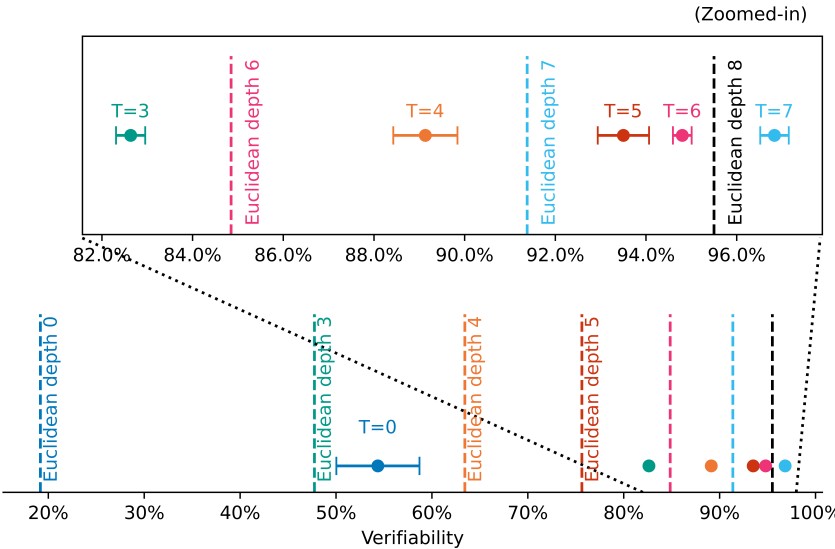

Figure 2: **Verifiability with increasing amounts of annotation**. $T$ is the number of steps added in Annotated Transcript Learning. Dashed lines indicate *Euclidean depth*, that bound the Verifiability of models that prove *only* for integers up to a certain number of steps. Each $T$ was run with three seeds, with mean $\pm$ standard error depicted. The upper graph provides a zoomed-in view of the 82% to 98% range from the lower graph, which spans a broader scale from 20% to 100%.

The proof $\pi$ is annotated by including intermediate steps in its computation. Details are deferred to Appendix E; roughly speaking, we observe that the proof $\pi$ for input $(\mathbf{a}, \mathbf{b})$ is obtained as the last element in a sequence $\mathbf{a}, \mathbf{b}, \pi_1, \pi_2, \ldots$ computed by the Euclidean algorithm. We annotate the proof $\pi$ by prepending to it the sequence of *Euclidean steps* $(\pi_1, \ldots, \pi_\mathbf{T})$ up to some fixed cutoff $T$.

Figure 2 shows how $T$ affects the Verifiability of the learned model. As suggested by Lee et al. (2024), training the model on more intermediate steps results in better performance; in our case,

---

[6]With probability 1, i.e., $s = 0$ in Definition 3.2.

increasing the number of intermediate steps $T$ yields better Self-Proving models. One might suspect that models only learn to execute the Euclidean algorithm in-context. To rule out this hypothesis, we derive an upper bound on the possible efficacy of such limited models. This bound is based on the *Euclidean depth* of integers $(x_1, x_2)$, which we define as the number of intermediate steps that the Euclidean algorithm makes before terminating on input $(x_1, x_2)$. Indeed, a model that only learns to compute (in-context) the simple arithmetic of the Euclidean algorithm would only be able to prove the correctness of inputs $(x_1, x_2)$ whose depth does not exceed the annotation cutoff $T$.

Figure 2 tells a different story: For each cutoff $T$, we estimate the probability that integers $x_1, x_2 \sim \mu$ have Euclidean depth at most $T$ on $10^5$ sampled pairs. Larger annotation cutoff $T$ increases Verifiability, but all models exceed their corresponding Euclidean depth bound.

### 5.3 BASE OF REPRESENTATION

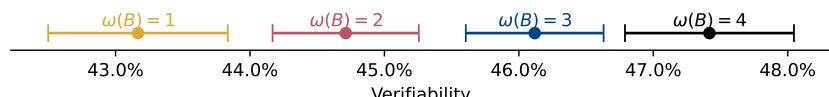

Figure 3: **The number of prime divisors of a base $\omega(B)$ determines Verifiability.** For each $o \in [4]$, we sampled 17 bases $B \in \{2, \ldots, 1386\}$ such that $\omega(B) = o$. A Self-Proving transformer was trained via Transcript Learning for twenty epochs on an identical dataset of 1024K samples encoded in base $B$. For each $\omega(B)$ we depict the mean $\pm$ standard error.

As mentioned previously, Charton (2024) concludes that, for a given base of representation $B$, transformers correctly compute the GCD of integers $x_1, x_2$ that are products of primes dividing $B$. Simply put, choosing a base $B$ with many different prime factors yields models with better correctness (accuracy), which suggests why base $B = 210 = 2 \cdot 3 \cdot 5 \cdot 7$ yielded the best results.

To test whether the factorization of $B$ has a similar effect on Verifiability as well, we train transformers on 68 bases varying the number of prime divisors $\omega(B)$ from $\omega(B) = 1$ (i.e., $B$ is a prime power) to $\omega(B) = 4$. Figure 3 shows that $\omega(B)$ correlates not just with correctness (Charton, 2024), but also with Verifiability. Although the finding is statistically significant (no overlapping error margins), the overall difference is by a few percentage points; we attribute this to the smaller (10%) number of samples on which models were trained, relative to our other experiments.

## 6 CONCLUSIONS

Trust between a learned model and its user is fundamental. In recent decades, Interactive Proofs (Goldwasser et al., 1985) have emerged as a general theory of trust established via verification algorithms. This work demonstrates that models can learn to formally prove their answers in an Interactive Proof system. We call models that possess this capability *Self-Proving*.

The definition of Self-Proving models forms a bridge between the rich theory of Interactive Proofs and the contemporary topic of Trustworthy ML. Interactive Proofs offer formal *worst-case soundness guarantees*; thus, users of Self-Proving models can be confident when their models generate correct answers—and detect incorrect answers with high probability.

We demonstrate the theoretical viability of our definition with two generic learning algorithms: Transcript Learning (TL) and Reinforcement Learning from Verifier Feedback (RLVF). The analyses of these algorithms is informed by techniques from theories of learning, RL, and computational complexity. This work can be extended in several directions: finding conditions for the convergence of RLVF, improving sample complexity bounds for TL, or designing altogether different learning algorithms (for example, by taking advantage of properties of the verifier).

To better understand the training dynamics of (Annotated) TL, we train Self-Proving transformers for the Greatest Common Divisor (GCD) problem. We train a small (6.3M parameter) transformer that learns to generate correct answers *and proofs* with high accuracy. Facing forward, we note that Interactive Proofs exist for capabilities far more complex than the GCD (Shamir, 1992); scaling up our experiments is the next step towards bringing Self-Proving models from theory to practice.

## ETHICS STATEMENT

This work proposes a theoretically-grounded approach to enhancing trust in learned models. By ensuring that models not only generate outputs but also prove their correctness to a verification algorithm, we tackle fundamental issues of trust and accountability in machine learning.

Self-Proving models build trust between models and users by offering formal worst-case soundness guarantees. This is particularly beneficial in high-stakes applications, such as healthcare and finance, where incorrect outputs can have severe consequences. The ability to verify correctness on a per-instance basis helps prevent potentially harmful decisions. It allows any user to decide for herself whether she trusts a particular output generated by the model, rather than relying on average-case guarantees (e.g., high scores on benchmarks as reported by the model's developer).

Furthermore, Self-Proving models promote accountability by allowing stakeholders to independently verify the correctness of a model's outputs. In particular, lawmakers and regulators could require models used in sensitive settings to be Self-Proving.

With that said, Self-Proving models also introduce challenges which must be addressed. First, we expect Self-Proving models to be harder to learn (in practice), which may limit their applicability in more complex tasks. Second, as with any learned model, Self-Proving models could be used in harmful ways; developers of a model (and verification algorithm) must consider the impact of their systems in the specific context in which they are deployed (Suresh et al., 2023). In other words, the fact that a Self-Proving model's outputs are provably correct does not mean that these outputs were ought to be generated in the first place.

## REPRODUCIBILITY STATEMENT

The pseudocode for Transcript Learning (TL) and Reinforcement Learning from Verifier Feedback (RLVF) is specified in Algorithms 1 and 2, respectively. Their implementation is available in the `self-proving-models` Python package; this package and all other code necessary to reproduce the experiments in Section 5 are attached as supplementary material, and will be released under the MIT license upon publication. The compute requirements, model architecture and hyperparameters are all detailed in Appendix F. Datasets and model checkpoints from the experiments in Section 5 are available via an anonymous link,[7] and will be made public upon publication.

As for the theoretical results in Section 4, the formal statement of assumptions and proofs can be found in Appendix B.

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

## LIMITATIONS

Our experiments are focused on a single ground-truth capability, namely, computing the GCD. Yet, the theoretical portion of our work holds for any ground-truth $F^*$ that admits an Interactive Proof system. Training large Self-Proving models for more complex ground-truths will likely pose additional practical learning challenges. With that said, we stress that generating accepting transcripts for use in Transcript Learning is distinct from these learning challenges. Collecting accepting transcripts is a purely computational task, and can even be done "offline" prior to the model's training.

Additionally, in our current learning methods, each individual ground-truth capability requires training a separate Self-Proving model. It would be interesting to adapt our definition and methods to deal with a single *generalist* Self-Proving model that proves its correctness to multiple verifiers of different ground-truths.

## A   A DEFINITION FOR GENERAL LOSS FUNCTIONS AND ONE-TO-MANY RELATIONS

We present a variant of Self-Proving models (Definition 3.4) generalized in two ways.

**General (bounded) loss functions.** In Definition 3.1 we implicitly use the 0-1 loss when measuring the correctness of a model: For any $x \in X$, we measure only whether the model generated the correct output $y = F^*(x)$, but not how "far" the generated $y$ was from $F^*(x)$. It is often the case in machine learning that we would be satisfied with models that generate a "nearly-correct" output. This is formalized by specifying a loss function $\ell \colon \Sigma^* \times \Sigma^* \to [0, 1]$ and measuring the probability that $\ell(x, y)$ is smaller than some threshold $\lambda \in [0, 1)$, where $x$ is drawn from the input distribution $\mu$, and $y$ is generated by the model when given input $x$.

In the context of language modeling, different loss function allow for a more fine-grained treatment of the *semantics* of a given task. As an example, consider the *prime-counting task*:

- Given an integer $x < 10^9$, output the number of primes less than or equal to $x$.

In the notation of Section 3, the prime-counting task would be captured by the ground-truth function

$$F^*(x) \coloneqq |\{p \in \mathbb{N} \mid p \leq x, \ p \text{ is prime}\}| .^8$$

Per Definition 3.1, any output other than $F^*(x)$ is "just as incorrect" as any other. Yet, we might prefer outputs that are closer to the correct answer, say, in $L_1$ norm. This preference can be captured by the following bounded loss function

$$\ell_1(x, y) \coloneqq \begin{cases} |y - F^*(x)| \cdot 10^{-9} & \text{if } y \leq 10^9 \\ 1 & \text{else.} \end{cases}$$

In particular, if we are interested in knowing the answer only up to some additive constant $C$, we could say that an output $y$ is "correct-enough" if $\ell_1(x, y) \leq C \cdot 10^{-9}$.

More generally, we relax Definition 3.1 to capture approximate correctness as follows.

**Definition A.1** (Approximate correctness). *Let $\mu$ be a distribution over input sequences in $\Sigma^*$ and let $\ell \colon \Sigma^* \times \Sigma^* \to [0, 1]$ be a loss function. For any $\alpha, \lambda \in [0, 1]$, we say that model $F_\theta$ is $(\alpha, \lambda)$-correct with respect to $\mu$ if*

$$\Pr_{\substack{x \sim \mu \\ y \sim F_\theta(x)}} [\ell(x, y) \leq \lambda] \geq \alpha.$$

**One-to-many-relations.** In Section 3, we focused on the setting of models of a ground-truth function $F^* \colon \Sigma^* \to \Sigma^*$. That is, when each input $x$ has exactly one correct output, namely $F^*(x)$. A more general setting would be to consider a ground-truth *relation* $L \subseteq \Sigma^* \times \Sigma^*$. Then, we say that $y$ is a correct output for $x$ if $(x, y) \in L$. Importantly, this allows a single $x$ to have many possible correct outputs, or none at all.

Note that we must take care to choose a loss function $\ell$ that captures correctness with respect to the relation $L$, i.e., $\ell(x, y) = 0$ if and only if $(x, y) \in L$. Equivalently, any loss function $\ell$ induces a relation $L \coloneqq \{(x, y) \mid \ell(x, y) = 0\}$. Therefore, our relaxation to approximate-correctness Definition A.1 already captures the setting of one-to-many relations, since an input $x$ may have multiple $y^*$ such that $\ell(x, y^*) = 0$.

### A.1 THE GENERAL DEFINITION

We first present a relaxed definition of Interactive Proof systems for verifying approximate-correctness.

**Definition A.2** (Definition 3.2, generalized). *Fix a soundness error $s \in (0, 1)$, a threshold $\lambda \in [0, 1)$, a finite set of tokens $\Sigma$, and a loss function $\ell \colon \Sigma^* \times \Sigma^* \to [0, 1]$. A verifier $V$ for $\ell$ with threshold $\lambda$ is a probabilistic polynomial-time algorithm that is given explicit inputs $x, y \in \Sigma^*$ and black-box (oracle) query access to a prover $P$. It interacts with $P$ over $R$ rounds (see Figure 1) and outputs a decision $\langle V, P \rangle (x, y) \in \{\text{reject}, \text{accept}\}$. Verifier $V$ satisfies the following two guarantees:*

- Completeness: *There exists an honest prover $P^*$ such that, for all $x, y \in \Sigma^*$, if $\ell(x, y) = 0$ then*

$$\Pr[\langle V, P^* \rangle (x, y) \text{ accepts}] = 1,$$

  *where the probability is over the randomness of $V$.*

---

[8] Formally, the input and output are strings in $\Sigma^*$ representing integers (e.g. in decimal representation). See Appendix F for a concrete instantiation used in our experiments.

- Soundness: *For all $P$ and for all $x, y \in \Sigma^*$, if $\ell(x, y) > \lambda$ then*

$$\Pr[\langle V, P \rangle (x, y) \text{ accepts}] \leq s,$$

   *where the probability is over the randomness of $V$ and $P$, and $s$ is the soundness error.*

Indeed, for a given ground-truth function $F^* \colon \Sigma^* \to \Sigma^*$, Definition 3.2 can be recovered by choosing the 0-1 loss

$$\ell_{F^*}(x, y) := \begin{cases} 1 & \text{if } x \neq F^*(y) \\ 0 & \text{else.} \end{cases}$$

and any threshold $\lambda \in [0, 1)$.

**Remark A.3** (Connection to Interactive Proofs of Proximity)**.** *Definition A.2 can be seen as a slight generalization of (perfect completeness) Interactive Proofs of Proximity (IPPs, Rothblum et al. 2013). An IPP for a relation $L \subseteq \Sigma^* \times \Sigma^*$ with proximity parameter $\lambda$ is obtained by instantiating Definition A.2 with the loss function $\ell_{\text{Hamming}}$ defined by*

$$\ell_{\text{Hamming}}(x, y) := \min \left\{ \frac{\#\{i \mid y_i \neq y_i^*\}}{|y|} \;\middle|\; (x, y^*) \in L, \; |y^*| = |y| \right\},$$

*that is, $\ell_{\text{Hamming}}(x, y)$ is the fraction of tokens in $y$ that must be changed so as obtain an output $y^*$ with $(x, y^*) \in L$. However, the motivation of Rothblum et al. (2013) was studying sublinear time verification, whereas ours is to relax the requirements of traditional Interactive Proofs towards meeting common desiderata in machine learning.*

With this relaxed notion of Interactive Proofs in hand, we are now ready to define Self-Proving models for general (bounded) loss functions.

**Definition A.4** (Definition 3.4, generalized)**.** *Fix a loss function $\ell \colon \Sigma^* \times \Sigma^* \to [0, 1]$, a verifier $V$ for $\ell$ with threshold $\lambda \in [0, 1)$ as in Definition A.2, and a distribution $\mu$ over inputs $\Sigma^*$. The* Verifiability *of a model $P_\theta := \Sigma^* \to \Sigma^*$ is defined as*

$$\text{ver}_{V,\mu}(\theta) := \Pr_{\substack{x \sim \mu \\ y \sim P_\theta(x)}} [\langle V, P_\theta \rangle (x, y) \text{ accepts}] .$$

*We say that model $P_\theta$ is $\beta$-Self-Proving with respect to $V$ and $\mu$ if $\text{ver}_{V,\mu}(\theta) \geq \beta$.*

Analogously to Remark 3.5, we observe that Verifiability (Definition A.4) implies approximate-correctness: Suppose $P_\theta$ is $\beta$-Self-Proving model with respect to a verifier $V$ that has soundness error $s$ and threshold parameter $\lambda$ for loss function $\ell$. Then by a union bound,

$$\Pr_{\substack{x \sim \mu \\ y \sim P_\theta(x)}} [\ell(x, y) \leq \lambda] \geq \beta - s.$$

Importantly, as emphasized throughout this paper, soundness of $V$ implies that for *all* inputs $x$, any output $y$ such that $\ell(x, y) > \lambda$ is rejected with high probability $(1 - s)$.

# B    THEORETICAL ANALYSES FOR SECTION 4

In this section we provide a formal description and analysis of Transcript Learning (TL, Section 4.1) and Reinforcement Learning from Verifier Feedback (RLVF, Section 4.2). In Appendix B.1 we prove a convergence theorem for TL under convexity and Lipschitzness assumptions. Obtaining an analogous result for RLVF is more challenging; in lieu of a full analysis, we provide a lemma showing that the gradients estimated in the algorithm approximate the Verifiability of the model in Appendix B.2.

**Specification of the learning model.**    We must first fully specify the theoretical framework in which our results reside. Continuing from Section 3, we define a *learner* as an algorithm $\Lambda$ with access to a family of autoregressive models $\{P_\theta\}_\theta$ and samples from the input distribution $x \sim \mu$. In our setting of Self-Proving models (and in consistence with the Interactive Proofs literature), we give the learner the full specification of the verifier $V$. More formally,

**Definition B.1** (Self-Proving model learner). *A (Self-Proving model) learner is a probabilistic oracle Turing Machine* $\Lambda$ *with the following access:*

- *A family of* autoregressive models $\{P_\theta\}_{\theta \in \mathbb{R}^d}$ *where* $d \in \mathbb{N}$ *is the number of parameters in the family. Recall (Section 4) that for each* $\theta$ *and* $z \in \Sigma^*$, *the random variable* $P_\theta(z)$ *is determined by the logits* $\log p_\theta(z) \in \mathbb{R}^{|\Sigma|}$. *For any* $z \in \Sigma^*$ *and* $\sigma \in \Sigma$, *the learner* $\Lambda$ *can compute the gradient of the* $\sigma^{th}$ *logit, that is,* $\nabla_\theta \log \Pr_{\sigma' \sim p_\theta(z)}[\sigma = \sigma']$. *In particular,* $\log \Pr_{\sigma' \sim p_\theta(z)}[\sigma = \sigma']$ *is always differentiable in* $\theta$.

- *Sample access to the* input distribution $\mu$. *That is,* $\Lambda$ *can sample* $x \sim \mu$.

- *The full specification of the verifier* $V$, *i.e., the ability to emulate the verification algorithm* $V$. *More specifically,* $\Lambda$ *is able to compute* $V$*'s decision after any given interaction; that is, given input* $x$, *output* $y$, *and a sequence of queries and answers* $(q_i, a_i)_{i=1}^R$, *the learner* $\Lambda$ *can compute the decision of* $V$ *after this interaction.*

Throughout this section, we will refer to the *transcript* of an interaction between a verifier and a prover (see Figure 1). We will denote this transcript by $\pi = (y, q_1, a_1, \ldots, q_R, a_R)$, and for any index $s \in [|\pi|]$ we will write $\pi_{<s} \in \Sigma^{s-1}$ to denote the $s$-long prefix of $\pi$.

### B.1 Transcript Learning

Recall that Transcript Learning requires access to an *honest transcript generator*. Before we can formally define this object, it will be useful to define a *query generator* for a verifier $V$.

**Definition B.2** (Query generator). *Fix a verifier* $V$ *in a proof system with* $R \in \mathbb{N}$ *rounds, where the verifier issues queries of length* $L_q = |q_i|$ *and the prover (model) responses with answers of length* $L_a = |a_i|$.[9] *The* query generator $V_q$ *corresponding to* $V$ *takes as input a partial interaction and samples from the distribution over next queries by* $V$. *Formally, for any* $r \leq R$, *given input* $x$, *output* $y$, *and partial interaction* $(q_i, a_i)_{i=1}^r$, $V_q(x, y, q_1, a_1, \ldots, q_r, a_r)$ *is a random variable over* $\Sigma^{L_q}$.[10]

A *transcript generator* is a random variable over transcripts that faithfully represents the interaction of the verifier with some prover for a given input. An *honest transcript generator* is one who is fully supported on transcripts accepted by the verifier. We denote accepting transcripts by $\pi^* = (y^*, q_1^*, a_1^*, \ldots, q_R^*, a_R^*)$.

**Definition B.3** (Transcript generator). *Fix a verifier* $V$ *in a proof system of* $R \in \mathbb{N}$ *rounds. A transcript generator* $\mathcal{T}_V$ *for* $V$ *is a randomized mapping from inputs* $x \in \Sigma^*$ *to transcripts* $\pi = (y, q_1, a_1, \ldots, q_R, a_R) \in \Sigma^*$. *For any input* $x$, $\mathcal{T}_V(x)$ *satisfies that for each* $r \leq R$, *the marginal of* $\mathcal{T}_V(x)$ *on the* $r^{th}$ *query* $q_r$ *agrees with the corresponding marginal of the query generator* $(V_q)_r$.

*A transcript generator* $\mathcal{T}_V^* := \mathcal{T}_V$ *is* honest *if it is fully supported on transcripts* $\pi^*$ *for which the verifier accepts.*

Notice that for any verifier $V$, there is a one-to-one correspondence between transcript generators and (possibly randomized) provers. We intentionally chose *not* to specify a prover in Definition B.3 to emphasize that transcripts can be "collected" independently of the honest prover (see completeness in Definition 3.2), and in fact can be collected "in advance" prior to learning (see Figure 4). As long as the generator is fully supported on honest transcripts, it can be used for Transcript Learning (Algorithm 1 described next).

Convergence of TL is proven by a reduction to Stochastic Gradient Descent (SGD). Essentially, we are tasked with proving that TL estimates a surrogate of the Verifiability-gradient of its model $P_\theta$. More precisely, TL estimates the gradient of a function that bounds the Verifiability from below. Maximizing this function therefore maximizes the Verifiability.

The lower-bounding function is the agreement of the answers generated by $P_\theta$ with the answers provided by the honest transcript generator $\mathcal{T}_V^*$. More formally, we let $\mathcal{T}_V^\theta$ denote the transcript generator induced by the model $P_\theta$ when interacting with $V$: for each $x$, $\mathcal{T}_V^\theta(x)$ is the distribution

---

[9]We can assume that queries (resp. answers) all have the same length by padding shorter ones.

[10]For completeness' sake, we can say that when prompted with any sequence $z$ that does not encode an interaction, $V_q(z)$ is fully supported on a dummy sequence $\perp \cdots \perp \in \Sigma^{L_q}$.

---

**Algorithm 1:** Transcript Learning (TL)

---

**Hyperparameters:** Learning rate $\lambda \in (0, 1)$ and number of samples $N \in \mathbb{N}$.
**Input:** An autoregressive model family $\{P_\theta\}_{\theta \in \mathbb{R}^d}$, verifier specification (code) $V$, and sample
access to an input distribution $\mu$ and an accepting transcript generator $\mathcal{T}_V^*(\cdot)$.
**Output:** A vector of parameters $\bar{\theta} \in \mathbb{R}^d$.

1 Initialize $\theta_0 := \vec{0}$.
2 **for** $i = 0, \ldots, N - 1$ **do**
3   Sample $x \sim \mu$ and $\pi^* = (y^*, q_1^*, a_1^*, \ldots, q_R^*, a_R^*) \sim \mathcal{T}_V^*(x)$. Denote $a_0 := y^*$.
4   **foreach** *Round of interaction* $r = 0, \ldots, R$ **do**
5     Let $S(r)$ denote the indices of the $r^{\text{th}}$ answer $a_r$ in $\pi^*$, and let $\pi_{<s}$ denote the prefix of
      the partial transcript $(y, q_1^*, a_1^*, \ldots, q_r^*)$.
6     **for** $s \in S(r)$ **do**
7       Compute                                    # Forwards and backwards pass

$$\alpha_s(\theta_i) := \Pr_{\sigma \sim p_{\theta_i}(x\pi_{<s})}[\sigma = \pi_s^*]$$

$$\vec{d}_s(\theta_i) := \nabla_\theta \log \alpha_s(\theta_i) = \nabla_\theta \log \Pr_{\sigma \sim p_{\theta_i}(x\pi_{<s})}[\sigma = \pi_s^*].$$

8   Update

$$\theta_{i+1} := \theta_i + \lambda \cdot \prod_{\substack{r \in [R] \cup \{0\} \\ s \in S(r)}} \alpha_s(\theta_i) \cdot \sum_{\substack{r \in [R] \cup \{0\} \\ s \in S(r)}} \vec{d}_s(\theta_i).$$

9 Output $\bar{\theta} := \frac{1}{N} \sum_{i \in [N]} \theta_i$.

---

over transcripts of interactions between $V$ and $P_\theta$ on input $x$. We stress that $\pi^* \sim \mathcal{T}_V^*(x)$ and $\pi \sim \mathcal{T}_V^\theta(x)$ are transcripts produced when interacting *with the same verifier queries*; we can think of the verifier as simultaneously interacting with the honest prover and with the model $P_\theta$.[11] In what follows, we use $\pi^* \sim \mathcal{T}_V^*(x)$ and $\pi \sim \mathcal{T}_V^\theta(x)$ to denote two transcripts that share the same queries. That is, taking $\pi^* = (y^*, q_1^*, a_1^*, \ldots, q_R^*, a_R^*)$ to denote an accepting transcript sampled from $\mathcal{T}_V^*(x)$, and $\pi = (y, q_1^*, a_1, \ldots, q_R^*, a_R)$ to denote a random transcript sampled from $\mathcal{T}_V^\theta(x)$, we say that $\pi$ and $\pi^*$ *agree* if they agree on the prover answers, namely if:

$$(y, a_1, \ldots, a_R) = (y^*, a_1^*, \ldots, a_R^*).$$

This definition implicitly uses the independence of the verifier and model's randomness. We first prove that TL correctly estimates the gradient of $A(\theta)$ in its update step.

**Lemma B.4** (TL gradient estimation). *Fix an input distribution $\mu$ over $\Sigma^*$ and a verifier $V$ with round complexity $R$ and answer length $L_a$. Fix an honest transcript generator $\mathcal{T}_V^*$. Let $\theta$ be the parameters of a model $P_\theta$ and let*

$$A(\theta) := \Pr_{\substack{x \sim \mu \\ \pi^* \sim \mathcal{T}_V^*(x) \\ \pi \sim \mathcal{T}_V^\theta(x)}} [\pi = \pi^*].$$

*Then,*

$$\nabla A(\theta) = \mathbb{E}_{\substack{x \sim \mu \\ \pi^* \sim \mathcal{T}_V^*}} \left[ \prod_{\substack{r \in [R] \cup \{0\} \\ s \in S(r)}} \alpha_s(\theta) \cdot \sum_{\substack{r \in [R] \cup \{0\} \\ s \in S(r)}} \vec{d}_s(\theta) \right]$$

*where $S(r)$, $\alpha_s(\theta)$ and $\vec{d}_s(\theta)$ are as defined in Algorithm 1.*

---

[11]The way it is presented in the algorithm (and implemented in the experiments), first the verifier is called by $\mathcal{T}_V^*$ and outputs queries $(q_1^*, \ldots q_R^*)$, and then the model is prompted with the verifier queries one a time. This maintains soundness, since a proof system is sound as long as the prover does not know the verifier's queries in advance.

Phase 1: Collect honest transcripts     Phase 2: Transcript Learning

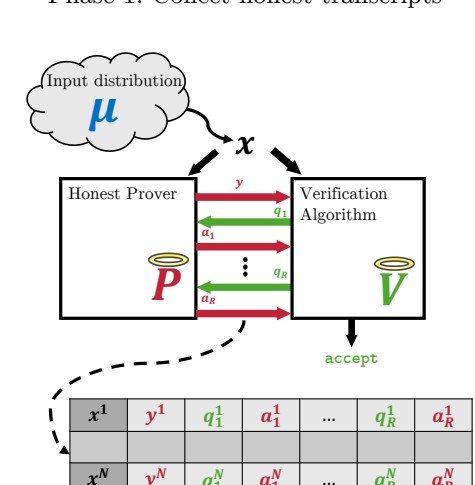

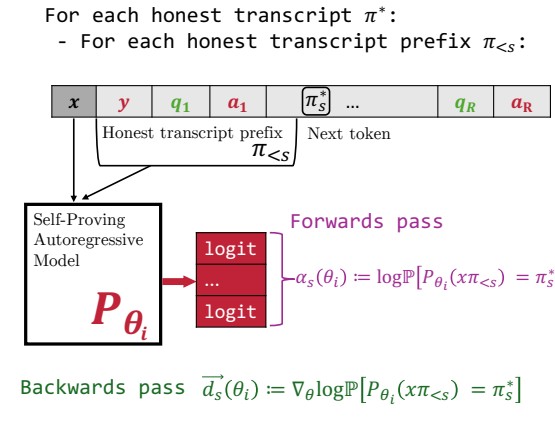

Figure 4: **Transcript Learning, visualized.** To understand Algorithm 1, consider the above visualization. In Phase 1, $N$ honest transcripts are collected by letting an Honest Prover interact with the Verification Algorithm; these will be the samples from the honest transcript generator $\mathcal{T}_V^*(x)$. Phase 2 describes the execution of Algorithm 1 itself: For each honest transcript $\pi^*$ (lines 2-3), and for each prefix $\pi_s$ of this transcript (lines 4-6), the $\alpha_s(\theta_i)$ and $\vec{d}_s(\theta_i)$ are computed via forwards and backwards passes, respectively (line 7). After iterating through all prefixes, the parameters $\theta_i$ are updated (line 8).

Note that Lemma B.4 is true for *any* model $P_\theta$. Moreover, the random vector over which the expectation is taken (in the right hand side) is precisely the direction of the update performed in Algorithm 1. We now prove Lemma B.4, from which we derive Theorem 4.1.

*Proof.* Throughout this proof, expectations and probabilities will be over the same distributions as in the lemma statement. First, we use the law of total probability together with the autoregressive property of $P_\theta$ (Section 4) to switch from probabilities on transcripts, to products of next-token probabilities. Formally, consider a fixed input $x$, an honest transcript $\pi^* = (y^*, q_1^*, a_1^*, \ldots, q_R^*, a_R^*)$, and denote a random transcript sampled from $\mathcal{T}_V^\theta(x)$ when using the same verifier queries by $\pi = (y, q_1^*, a_1, \ldots, q_R^*, a_R)$. For any $r \in [R]$ denote the random variable $\mathcal{T}_V^{\theta, <r} := \mathcal{T}_V^\theta(y q_1^* a_1 \cdots a_{r-1} q_r^*)$. Then,

$$\Pr_\pi[\pi = \pi^*] = \Pr_\pi[(y, a_1, \ldots, a_R) = (y^*, a_1^*, \ldots, a_R^*)] \tag{3}$$

$$= \Pr_{y \sim P_\theta(x)}[y = y^*] \cdot \prod_{r \in [R]} \Pr_{a \sim \mathcal{T}_V^{\theta, <r}}[a = a_r^*]$$

$$= \Pr_{y \sim P_\theta(x)}[y = y^*] \cdot \prod_{\substack{r \in [R] \\ s \in S(r)}} \Pr_{\sigma \sim p_\theta(\pi_{<s}^*)}[\sigma = \pi_s^*] \tag{4}$$

$$= \prod_{\substack{r \in [R] \cup \{0\} \\ s \in S(r)}} \alpha_s(\theta), \tag{5}$$

where, as noted above, Equation (3) uses the independence of the verifier and model's randomness, Equation (4) uses the autoregressive property of $P_\theta$ (Definition B.1), and Equation (5) is by definition

of $\alpha_s$ and of $a_0$. Next, a basic calculus identity gives

$$\nabla_\theta \left( \Pr_\pi [\pi = \pi^*] \right) = \Pr_\pi [\pi = \pi^*] \cdot \nabla_\theta \log \left( \Pr_\pi [\pi = \pi^*] \right). \tag{6}$$

This implicitly assumes that $\Pr_\pi [\pi = \pi^*]$ is differentiable in $\theta$; indeed, this follows from Definition B.1, where the logits of the model were assumed to by differentiable. Let us focus on the rightmost factor. By Equation (5),

$$\nabla_\theta \log \left( \Pr_\pi [\pi = \pi^*] \right) = \nabla_\theta \log \left( \prod_{\substack{r \in [R] \cup \{0\} \\ s \in S(r)}} \alpha_s(\theta) \right) = \sum_{\substack{r \in [R] \cup \{0\} \\ s \in S(r)}} \nabla_\theta \log \alpha_s(\theta) = \sum_{\substack{r \in [R] \cup \{0\} \\ s \in S(r)}} \vec{d}_s(\theta) \tag{7}$$

where the last equality is by definition of $\vec{d}_s(\theta)$. Combining Equation (5) and Equation (6) gives

$$\nabla_\theta \left( \Pr_\pi [\pi = \pi^*] \right) = \prod_{\substack{r \in [R] \cup \{0\} \\ s \in S(r)}} \alpha_s(\theta) \cdot \sum_{\substack{r \in [R] \cup \{0\} \\ s \in S(r)}} \vec{d}_s(\theta).$$

By the law of total probability and the linearity of the gradient,

$$\mathbb{E}_{x,\pi^*} \left[ \nabla_\theta \left( \Pr_\pi [\pi = \pi^*] \right) \right] = \nabla_\theta \left( \mathbb{E}_{x,\pi^*} \left[ \Pr_\pi [\pi = \pi^*] \right] \right) = \nabla_\theta \left( \Pr_{x,\pi^*,\pi} [\pi = \pi^*] \right) = \nabla_\theta A(\theta).$$

which concludes the proof. $\qquad\square$

We are now ready to prove Theorem 4.1. We restate it below in full formality.

**Theorem B.5** (Theorem 4.1, formal)**.** *Fix a verifier $V$, an input distribution $\mu$, an autoregressive model family $\{P_\theta\}_{\theta \in \mathbb{R}^d}$, and a norm $|| \cdot ||$ on $\mathbb{R}^d$. Fix an honest transcript generator $\mathcal{T}_V^*$, and assume that the* agreement function

$$A(\theta) \coloneqq \Pr_{\substack{x \sim \mu \\ \pi^* \sim \mathcal{T}_V^*(x) \\ \pi \sim \mathcal{T}_V^\theta(x)}} [\pi = \pi^*]$$

*is concave in $\theta$, where the verifier queries are the same in $\pi^*$ and $\pi$. For any $\varepsilon > 0$, let $B_{\mathrm{Norm}}$, $B_{\mathrm{Lip}}$ and $C$ be upper-bounds such that the following conditions hold.*

- *There exists $\theta^* \in \mathbb{R}^d$ with $||\theta^*|| < B_{\mathrm{Norm}}$ such that $A(\theta^*) \geq 1 - \varepsilon/2$.*

- *For all $\theta$, the logits of $P_\theta$ are $B_{\mathrm{Lip}}$-Lipschitz in $\theta$. That is,*

$$\sup_{\substack{\theta \in \mathbb{R}^d \\ z \in \Sigma^*}} ||\nabla_\theta \log p_\theta(z)|| \leq B_{\mathrm{Lip}}.$$

- *In the proof system defined by $V$, the total number of tokens (over all rounds) is at most $C$.*

*Denote by $\bar{\theta}$ the output of TL running for number of iterations $N$ where*

$$N \geq 4 \cdot C^2 \cdot \frac{B_{\mathrm{Norm}}^2 \cdot B_{\mathrm{Lip}}^2}{\varepsilon^2}$$

*and learning rate $\lambda = B_{\mathrm{Norm}}/C B_{\mathrm{Lip}}\sqrt{N}$. Then the expected Verifiability (over the randomness of the samples collected by TL) of $\bar{\theta}$ is at least $1 - \varepsilon$. That is,*

$$\mathbb{E}_{\bar{\theta}}[\mathrm{ver}_{V,\mu}(\bar{\theta})] \geq 1 - \varepsilon.$$

*Proof.* Our strategy is to cast TL as Stochastic Gradient Ascent and apply Fact C.2. Let $\varepsilon$, $B_{\mathrm{Norm}}$, $B_{\mathrm{Lip}}$ and $C$ as in the theorem statement be given. Let $\theta^*$ be such that $A(\theta^*) \geq 1 - \varepsilon/2$ and $||\theta^*|| \leq B_{\mathrm{Norm}}$.

First, notice that

$$\mathbb{E}_{\bar{\theta}}\left[\mathrm{ver}_{V,\mu}(\bar{\theta})\right] \geq \mathbb{E}_{\bar{\theta}}[A(\bar{\theta})],$$

This is because, for any $x$ and model $P_\theta$, whenever the transcript generated by $\mathcal{T}^\theta(x)$ agrees with $\pi^*$, then the verifier accepts (because $\pi^*$ is honest). Therefore, to prove the theorem it suffices to show that

$$\mathbb{E}_{\bar{\theta}}[A(\bar{\theta})] \geq 1 - \varepsilon.$$

Following the notation in Algorithm 1, in every iteration $i \in [N]$ the norm of the update step is

$$\left\| \prod_{\substack{r\in[R]\cup\{0\}\\ s\in S(r)}} \alpha_s(\theta_i) \cdot \sum_{\substack{r\in[R]\cup\{0\}\\ s\in S(r)}} \vec{d}_s(\theta_i) \right\| = \left\| \prod_{\substack{r\in[R]\cup\{0\}\\ s\in S(r)}} \alpha_s(\theta_i) \right\| \cdot \left\| \sum_{\substack{r\in[R]\cup\{0\}\\ s\in S(r)}} \vec{d}_s(\theta_i) \right\|$$

$$\leq 1 \cdot \sum_{\substack{r\in[R]\cup\{0\}\\ s\in S(r)}} \left\| \vec{d}_s(\theta_i) \right\|,$$

where the inequality is because $\alpha_s(\theta_i)$ are probabilities, so $\leq 1$. Continuing, we have

$$\sum_{\substack{r\in[R]\cup\{0\}\\ s\in S(r)}} \left\| \vec{d}_s(\theta_i) \right\| \leq \sum_{\substack{r\in[R]\cup\{0\}\\ s\in S(r)}} B_{\mathrm{Lip}} \leq C \cdot B_{\mathrm{Lip}}.$$

The first inequality is by definition of $B_{\mathrm{Lip}}$ as an upper-bound on the gradient of $P_\theta$'s logits. The second is because, by definition, $C$ is an upper-bound on the number of tokens sent by the prover in the proof system, which is exactly the number of terms in the sum: $r$ indexes rounds, and $s$ indexes tokens sent in each round.

To conclude, Lemma B.4 shows that TL samples from a gradient estimator for $A(\theta)$, while the above equation shows that the gradient is upper-bounded by $C \cdot B_{\mathrm{Lip}}$. We can therefore apply Fact C.2 to obtain

$$\mathbb{E}_{\bar{\theta}}\left[A\left(\bar{\theta}\right)\right] \geq A(\theta^*) - \varepsilon/2 \geq (1 - \varepsilon/2) - \varepsilon/2 = 1 - \varepsilon,$$

where the inequality is by definition of $\theta^*$.

$\square$

**Remark B.6** (On the realizability assumption in Theorem B.5). *The first condition in Theorem B.5 expresses a fundamental constraint: if a Self-Proving model cannot be realized within the chosen architecture, then learning such a model is impossible regardless of the training approach. Rather than being a limitation that requires justification, this represents a necessary logical precondition.*

*The challenge then lies in selecting an architecture capable of expressing a Prover for a given Proof System. One common approach assumes deep neural networks as universal function approximators, scaling both architecture size and training data until achieving desired performance. Recent theoretical work has established rigorous foundations for this approach, demonstrating the Turing-completeness of transformers (Bhattamishra et al., 2020) and their variants (Dehghani et al., 2019). These architectures can even approximate arbitrary continuous sequence-to-sequence functions on compact domains (Yun et al., 2020). Therefore, transformer architectures can realize any Turing machine—including the Prover in an Interactive Proof system, which operates within polynomial space bounds (or better: Goldwasser et al. 2015).*

### B.2 REINFORCEMENT LEARNING FROM VERIFIER FEEDBACK

Our second learning method, Reinforcement Learning from Verifier Feedback (RLVF, Algorithm 2), does not require access to an honest transcript generator. Instead, the learner generates transcripts herself by emulating the interaction of the verifier with the current Self-Proving model $P_\theta$. When an accepting transcript is generated, the learner updates the parameters $\theta$ towards generating such transcript.

---

**Algorithm 2:** Reinforcement Learning from Verifier Feedback (RLVF)

---

**Hyperparameters:** Learning rate $\lambda \in (0, 1)$ and number of samples $N \in \mathbb{N}$.
**Input:** An autoregressive model family $\{P_\theta\}_{\theta \in \mathbb{R}^d}$, initial parameters $\theta_0 \in \mathbb{R}^d$, verifier
      specification (code) $V$, and sample access to an input distribution $\mu$.
**Output:** A vector of parameters $\bar{\theta} \in \mathbb{R}^d$.

1 **for** $i = 0, \ldots, N - 1$ **do**
2     Sample $x \sim \mu$.
3     Initialize $a_0 \coloneqq y \sim P_{\theta_i}(x)$.
4     **foreach** *Round of interaction* $r = 1, \ldots R$ **do**
5         Sample the $r^{\text{th}}$ query                     # Emulate the verifier

$$q_r \sim V_q(x, a_0, q_1, a_1, \ldots, q_{r-1}, a_{r-1}).$$

        Sample the $r^{\text{th}}$ answer                      # Forwards pass

$$a_r \sim P_{\theta_i}(x, a_0, q_1, a_1, \ldots, q_r).$$

        Let $\tau_r \coloneqq (a_0, q_1, \ldots, a_{r-1}, q_r)$.
6         **for** $s \in [L_a]$ **do**
7             Let $a_{r,s}$ denote the $s^{\text{th}}$ token in $a_r$. Compute      # Backwards pass

$$\vec{d}_s(\theta_i) \coloneqq \nabla_\theta \log \Pr_{\sigma \sim p_{\theta_i}(x\tau_r)}[\sigma = a_{r,s}].$$

8     **if** $V(x, y, q_1, a_1, \ldots, q_R, a_R)$ accepts **then**
9         Update

$$\theta_{i+1} \coloneqq \theta_i + \lambda \cdot \sum_{\substack{r \in [R] \cup \{0\} \\ s \in [L_a]}} \vec{d}_s(\theta_i).$$

10 Output $\bar{\theta} \coloneqq \frac{1}{N} \sum_{i \in [N]} \theta_i$.

---

Before we continue with formal analysis of Algorithm 2, let us make a few observations.

Firstly, the parameters are updated (line 11) only when an accepting transcript was generated. This means that the learner can first fully generate the transcript (lines 6-7), and then take backwards passes (line 9) only if the transcript was accepted by $V$. This is useful in practice (e.g. when using neural models) as backwards passes are more computationally expensive than forwards passes.

On the other hand, this means that RLVF requires the parameter initialization $\theta_0$ to have Verifiability bounded away from 0, so that accepting transcripts are sampled with sufficient probability. Fortunately, such a Self-Proving base model can be learned using TL. This gives a learning paradigm in which a somewhat-Self-Proving base model is learned with TL (with Verifiability $\delta > 0$), and then "amplified" to a fully Self-Proving model using RLVF. This can be seen as an adaptation of the method of Nair et al. (2018) to the setting of Self-Proving models.

Secondly, in comparing Algorithms 1 and 2, we see that the latter (RLVF) does not keep track of the probabilities $\alpha_s$. This is because, in RL terms, RLVF is an *on-policy* algorithm; it generates transcripts using the current learned model, unlike TL that samples them from a distribution whose parameterization is unknown to the learner. Hence, the update step in RLVF is simpler than TL.

We now prove that the update step in RLVF maximizes the Verifiability of $P_\theta$; this is analogous to Lemma B.4 for TL. We leave it for future work to use Lemma B.7 to obtain convergence bounds on RLVF (analogous to Theorem B.5). As mentioned in Section 4.2, the gap between the lemma and a full convergence theorem (informally) reduces to the problem of obtaining convergence bounds for Policy Gradient methods, a challenging and active research direction (e.g. Agarwal et al. 2021).

Indeed, the update step that the algorithm takes can be expressed as the random vector over which the expectation is taken (in the right hand side).

**Lemma B.7** (RLVF gradient estimation). *Fix an input distribution $\mu$ over $\Sigma^*$ and a verifier $V$ with round complexity $R$ and answer length $L_a$. For any transcript $(x, y, q_1, \ldots, a_R)$ we let*

---

$\mathrm{Acc}_V(x, y, q_1, \ldots, a_R)$ *denote the indicator random variable which equals 1 if and only if $V$ accepts the transcript. For any model $P_\theta$, denote by* $\mathrm{ver}(\theta)$ *the verifiability of $P_\theta$ with respect to $V$ and $\mu$ (Definition 3.4). Then, for any $\theta$,*

$$\nabla_\theta \mathrm{ver}(\theta) = \mathop{\mathbb{E}}_{\substack{x \sim \mu \\ y \sim P_\theta(x) \\ (q_r, a_r)_{r=1}^R}} \left[ \mathrm{Acc}_V(x, y, q_1, \ldots, a_R) \cdot \sum_{\substack{r \in [R] \cup \{0\} \\ s \in [L_a]}} \vec{d}_s(\theta) \right]$$

*where $(q_r, a_r)_{r=1}^R$ are as sampled in lines 5-6 of Algorithm 2, and $\vec{d}_s(\theta)$ is as defined in line 8 therein.*

*Proof.* Recall the *transcript generator of $P_\theta$*, denoted by $\mathcal{T}_V^\theta$ (see Lemma B.4). By the definitions of Verifiability in Definition 3.4 and $V(x, y, q_1, \ldots, a_R)$ in the lemma statement,

$$\begin{aligned}
\mathrm{ver}(\theta) &:= \mathop{\mathrm{Pr}}_{\substack{x \sim \mu \\ y \sim P_\theta(x)}} \left[ \langle V, P_\theta \rangle (x, y) \text{ accepts} \right] \\
&= \mathop{\mathbb{E}}_{\substack{x \sim \mu \\ y \sim P_\theta(x) \\ (q_r, a_r)_{r=1}^R}} \left[ \mathrm{Acc}_V(x, y, q_1, \ldots, a_R) \right] \\
&= \mathop{\mathbb{E}}_{x \sim \mu} \left[ \mathop{\mathrm{Pr}}_{\pi \sim \mathcal{T}_V^\theta(x)} \left[ \mathrm{Acc}_V(x, \pi) \right] \right]
\end{aligned} \tag{8}$$

Now, for every input $x$, let $\Pi^*(x) \subset \Sigma^*$ denote the set of accepting transcripts:

$$\Pi^*(x) := \left\{ \pi^* \in \Sigma^* : \mathrm{Acc}_V(x, \pi^*) = 1 \right\}.$$

We can assume that $\Pi^*(x)$ has finite cardinality, since $V$'s running time is bounded and hence the number of different transcripts that it can read (and accept) is finite. For any fixed input $x$, we can express its acceptance probability by the finite sum:

$$\mathop{\mathrm{Pr}}_{\pi \sim \mathcal{T}_V^\theta(x)} \left[ \mathrm{Acc}_V(x, \pi) \right] = \sum_{\pi^* \in \Pi^*(x)} \mathop{\mathrm{Pr}}_{\pi \sim \mathcal{T}_V^\theta(x)} \left[ \pi = \pi^* \right]. \tag{9}$$

We will use Equations (3) through (7) in the proof of Lemma B.4. Up to a change in index notation, these show that, for any $\pi^*$,

$$\nabla_\theta \mathop{\mathrm{Pr}}_{\pi \sim \mathcal{T}^\theta(x)} \left[ \pi = \pi^* \right] = \mathop{\mathrm{Pr}}_{\pi \sim \mathcal{T}^\theta(x)} \left[ \pi = \pi^* \right] \cdot \sum_{\substack{r \in R \cup \{0\} \\ s \in [L_a]}} \nabla_\theta \vec{d}_s(\theta).$$

Combining Equations (8) and (9), by linearity of expectation we have that

$$\nabla_\theta \mathrm{ver}(\theta) = \mathop{\mathbb{E}}_{x\sim\mu}\left[\sum_{\pi^*\in\Pi^*(x)} \nabla_\theta \mathop{\Pr}_{\pi\sim\mathcal{T}^\theta(x)}[\pi = \pi^*]\right]$$

$$= \mathop{\mathbb{E}}_{x\sim\mu}\left[\sum_{\pi^*\in\Pi^*(x)} \mathop{\Pr}_{\pi\sim\mathcal{T}^\theta(x)}[\pi = \pi^*]\cdot \sum_{\substack{r\in R\cup\{0\}\\ s\in[L_a]}} \nabla_\theta\vec{d}_s(\theta)\right]$$

$$= \mathop{\mathbb{E}}_{x\sim\mu}\left[\mathop{\mathbb{E}}_{\pi\sim\mathcal{T}^\theta(x)}\left[\mathrm{Acc}_V(x,\pi)\cdot \sum_{\substack{r\in R\cup\{0\}\\ s\in[L_a]}} \nabla_\theta\vec{d}_s(\theta)\right]\right]$$

$$= \mathop{\mathbb{E}}_{\substack{x\sim\mu\\ \pi\sim\mathcal{T}^\theta(x)}}\left[\mathrm{Acc}_V(x,\pi)\cdot \sum_{\substack{r\in R\cup\{0\}\\ s\in[L_a]}} \nabla_\theta\vec{d}_s(\theta)\right]$$

$$= \mathop{\mathbb{E}}_{\substack{x\sim\mu\\ y\sim P_\theta(x)\\ (q_r,a_r)_{r=1}^R}}\left[\mathrm{Acc}_V(x,y,q_1,\ldots,a_R)\cdot \sum_{\substack{r\in R\cup\{0\}\\ s\in[L_a]}} \nabla_\theta\vec{d}_s(\theta)\right],$$

where in the last equality, the probability is over $(q_r, a_r)$ sampled as in Algorithm 2, and it follows from the definition of the transcript generator $\mathcal{T}^\theta(x)$. □

## C  Preliminaries on Stochastic Gradient Ascent

For convenience of the reader, we provide a description of Stochastic Gradient Ascent and quote a theorem on its convergence. We adapt the presentation in Shalev-Shwartz & Ben-David (2014), noting that they present Stochastic Gradient Descent in its more general form for non-differentiable unbounded functions.

Stochastic Gradient Ascent (SGA) is a fundamental technique in concave optimization. Given a concave function $f\colon \mathbb{R}^d \to [0,1]$, SGA starts at $w_0 = \vec{0} \in \mathbb{R}^d$ and tries to maximize $f(w)$ by taking a series of "steps." Than directly differentiating $f$, SGA instead relies on an estimation $\nabla f(w)$: in each iteration, SGA takes a step in a direction that estimates $\nabla f(w)$.

**Definition C.1** (Gradient estimator). *Fix a differentiable function $f\colon \mathbb{R}^d \to \mathbb{R}$ for some d. A* gradient estimator *for f is a randomized mapping $D_f\colon \mathbb{R}^d \to \mathbb{R}^d$ whose expectation is the gradient of f. That is, for all $w \in \mathbb{R}^d$,*

$$\mathop{\mathbb{E}}_{v\sim D_f(w)}[v] = \nabla f(w).$$

*Note that this is an equality between d-dimensional vectors.*

---

**Algorithm 3:** Stochastic Gradient Ascent

**Hyperparameters:** Learning rate $\lambda > 0$ and number of iterations $N \in \mathbb{N}$.
**Input:** A function $f\colon \mathbb{R}^d \to \mathbb{R}$ to maximize and a gradient estimator $D_f$ for $f$.
**Output:** A vector $\bar{w} \in \mathbb{R}^d$.
1 Initialize $w_0 := \vec{0} \in \mathbb{R}^d$.
2 **for** $i = 1, \ldots, N-1$ **do**
3     Sample $v_i \sim D_f(w_{i-1})$.
4     Update $w_i := w_{i-1} + \lambda \cdot v_i$.
5 Output $\bar{w} := \frac{1}{N}\sum_{i\in[N]} w_i$.

---

Theorem 14.8 in Shalev-Shwartz & Ben-David (2014) implies the following fact.

**Fact C.2.** *Fix a concave $f \colon \mathbb{R}^d \to [0, 1]$, a norm $|| \cdot ||$ on $\mathbb{R}^d$, and upper-bounds $B_{\mathrm{Norm}}, B_{\mathrm{Lip}} > 0$. Let*

$$w^* \in \underset{w \colon ||w|| < B_{\mathrm{Norm}}}{\mathrm{argmax}} \quad f(w),$$

*and let $\bar{w}$ denote the output of Algorithm 3 run for $N$ iterations with learning rate*

$$\lambda = \frac{B_{\mathrm{Norm}}}{B_{\mathrm{Lip}}\sqrt{N}}.$$

*If at every iteration it holds that $||v_i|| < B_{\mathrm{Lip}}$, then*

$$\underset{\bar{w}}{\mathbb{E}}\left[f(\bar{w})\right] \geq f(w^*) - \frac{B_{\mathrm{Norm}} \cdot B_{\mathrm{Lip}}}{\sqrt{N}}.$$

### C.1 LEARNING WITH STOCHASTIC GRADIENT ASCENT/DESCENT

Fact C.2 captures the general case of using SGA for maximization of concave problems. It is more common for the literature to discuss the equivalent setting of Stochastic Gradient Descent (SGD) for minimization of convex problems. Specifically, a common application of SGD is for the task of *Risk Minimization*: given a loss function and access to an unknown distribution of inputs, the goal is to minimize the expected loss with respect to the distribution. Assuming that the loss function is differentiable, the gradient of the loss serves as a gradient estimator (see Definition C.1) for the risk function. We refer the reader to Shalev-Shwartz & Ben-David (2014, Section 14.5.1) for a complete overview of SGD for risk minimization.

For the sake of completeness, we formulate Transcript Learning (TL, Algorithm 1) in the framework of Risk Minimization for Supervised Learning. Although multiple loss functions may achieve our ultimate goal—learning Self-Proving models—in what follows we define the loss that corresponds to TL. Fix a verifier $V$ and let $\mathcal{T}_V^*$ denote a distribution over accepting transcripts. We define

$$\mathsf{loss}\left(\theta, (x, \pi^*)\right) \coloneqq \underset{\pi \sim \mathcal{T}_V^\theta(x)}{\mathrm{Pr}}\left[\pi \neq \pi^*\right], \tag{10}$$

where $\pi^*$ and $\pi$ share the same verifier messages (as in Lemma B.4) so the inequality is only over the prover's messages, namely $\mathrm{Pr}_{\pi \sim \mathcal{T}_V^\theta(x)}\left[\pi \neq \pi^*\right] = \mathrm{Pr}_{\pi \sim \mathcal{T}_V^\theta(x)}[(y, a_1, \ldots, a_R) \neq (y^*, a_1^*, \ldots, a_R^*)].$[12]

The risk function is the expected value of the loss over the joint distribution of inputs and accepting transcripts $\mu \times \mathcal{T}_V^*(\mu)$:

$$\mathsf{Risk}\left(\theta\right) \coloneqq \underset{\substack{x \sim \mu \\ \pi^* \sim \mathcal{T}_V^*}}{\mathbb{E}}\left[\mathsf{loss}\left(\theta, (x, \pi^*)\right)\right],$$

which means that the *agreement function* defined in Theorem B.5

$$A(\theta) = \underset{\substack{x \sim \mu \\ \pi^* \sim \mathcal{T}_V^*(x) \\ \pi \sim \mathcal{T}_V^\theta(x)}}{\mathrm{Pr}}\left[\pi = \pi^*\right]$$

satisfies $A(\theta) = 1 - \mathsf{Risk}(\theta)$.

Thus, maximizing the agreement is equivalent to minimizing the risk. The hypothesis class over which the optimization is performed is the ball of radius $B_{\mathrm{Norm}}$, i.e., $\left\{\theta \in \mathbb{R}^d : ||\theta|| < B_{\mathrm{Norm}}\right\}$. The assumption that $A$ is concave in $\theta$ implies that the loss function is convex in $\theta$, which is the required assumption for using SGD for risk minimization.

Indeed, TL uses the natural gradient estimator for this setting, the gradient of the "complement" of the loss: $\mathrm{Pr}_\pi\left[\pi = \pi^*\right]$, since TL maximizes the agreement instead of minimizing the risk. The proof of Lemma B.4, i.e., $\nabla_\theta A(\theta) = \mathbb{E}_{x, \pi^*}\left[\nabla_\theta\left(\mathrm{Pr}_\pi\left[\pi = \pi^*\right]\right)\right]$, follows from the above discussion.

---

[12]This loss is not to be confused with those discussed in Appendix A. Here, we are simply explaining how TL can be viewed as a supervised risk minimizer for the loss function defined in Equation (10).

# D  ANNOTATIONS

We formally capture the modification described in Section 4.3 by introducing a *transcript annotator* and an *answer extractor* incorporated into the training and inference stages, respectively.

Fix a verifier $V$ in an $R$-round proof system with question length $L_q$ and answer length $L_a$. An *annotation system* with annotation length $\widetilde{L_a}$ consists of a *transcript annotator* $A$, and an *answer extractor* $E$.

In terms of efficiency, think of the annotator as an algorithm of the same computational resources as an honest prover in the system (see Definition 3.2), and the answer extractor as an extremely simple algorithm (e.g., trim a fixed amount of tokens from the annotation).

To use an annotation system the following changes need to be made:

- At training time, an input $x$ and transcript $\pi$ is annotated to obtain $\widetilde{\pi} := A(x, \pi)$, e.g. before the forwards backwards pass in TL (line 3 in Algorithm 1).

- At inference time (i.e., during interaction between $V$ and $P_\theta$), the prover keeps track of the annotated transcript, but in each round passes the model-generated (annotated) answer through the extractor $E$ before it is sent to the verifier. That is, in each round $r \in [R]$, the prover samples

$$\widetilde{a_r} \sim P_\theta(x, y, q_1, \widetilde{a_1}, \ldots, \widetilde{a_{r-1}}, q_r).$$

The prover then extracts an answer $a_r := E(\widetilde{a_r})$ which is sent to the verifier.

# E  A SIMPLE PROOF SYSTEM FOR THE GCD

The Euclidean algorithm for computing the Greatest Common Divisor (GCD) of two integers is possibly the oldest algorithm still in use today (Knuth, 1969). Its extended variant gives a simple proof system.

Before we dive in, let us clarify what we mean by *a proof system for the GCD*. Prover Paul has two integers 212 and 159; he claims that $GCD(212, 159) = 53$. An inefficient way for Verifier Veronica to check Paul's answer is by executing the Euclidean algorithm on $(212, 159)$ and confirm that the output is 53. In an efficient proof system, Veronica asks Paul for a short string $\pi^*$ (describing two integers) with which she can easily compute the answer—without having to repeat Paul's work all over. On the other hand, if Paul were to claim that "$GCD(212, 159) = 51$" (it does not), then for any alleged proof $\pi$, Veronica would detect an error and reject Paul's claim.

The verifier in the proof system relies on the following fact.

**Claim E.1** (Bézout's identity (Bezout, 1779)). *Let $x_0, x_1 \in \mathbb{N}$ and $z_0, z_1 \in \mathbb{Z}$. If $z_0 \cdot x_0 + z_1 \cdot x_1$ divides both $x_0$ and $x_1$, then $z_0 \cdot x_0 + z_1 \cdot x_1 = GCD(x_0, x_1)$.*

Any coefficients $z_0, z_1$ satisfying the assumption of Claim E.1 are known as *Bézout coefficients* for $(x_0, x_1)$. Claim E.1 immediately gives our simple proof system: For input $x = (x_0, x_1)$ and alleged GCD $y$, the honest prover sends (alleged) Bézout coefficients $(z_0, z_1)$. The Verifier accepts if and only if $y = z_0 \cdot x_0 + z_1 \cdot x_1$ and $y$ divides both $x_0$ and $x_1$.

In this proof system the Verifier does not need to make any query; to fit within Definition 3.2, we can have the verifier issue a dummy query. Furthermore, by Claim E.1 it is complete and has soundness error $s = 0$. Lastly, we note that the Verifier only needs to perform two multiplications, an addition, and two modulus operations; in that sense, verification is more efficient than computing the GCD in the Euclidean algorithm as required by Remark 3.3.

**Annotations.**  To describe how a proof $z = (z_0, z_1)$ is annotated, let us first note how it can be computed. The Bézout coefficients can be found by an extension of the Euclidean algorithm. It is described in Algorithm 4.[13]

---

[13]Our description follows https://en.wikipedia.org/wiki/Extended_Euclidean_algorithm.

---

**Algorithm 4:** Extended Euclidean algorithm

---

**Input:** Nonzero integers $x_0, x_1 \in \mathbb{N}$.
**Output:** Integers $(y, z_0, z_1)$, such that $y = GCD(x_0, x_1)$ and $(z_0, z_1)$ are Bézout coefficients
        for $(x_0, x_1)$.

1   Initialize $r_0 = x_0$, $r_1 = x_1$, $s_0 = 1$, $s_1 = 0$, and $q = 0$.
2   **while** $r_1 \neq 0$ **do**
3       Update $q := \lfloor r_0/r_1 \rfloor$.
4       Update $(r_0, r_1) := (r_1, r_0 - q \times r_1)$.
5       Update $(s_0, s_1) := (s_1, s_0 - q \times s_1)$.
6   Output GCD $y = r_0$ and Bézout coefficients $z_0 := s_0$ and $z_1 := (r_0 - s_0 \cdot x_0)/x_1$.

---

Referring to Algorithm 4, the annotation of a proof $z = (z_0, z_1)$ will consist of intermediate steps in its computation. Suppose that in each iteration of the While-loop, the algorithm stores each of $r_0$, $s_0$ and $q$ in an arrays $\vec{r_0}$, $\vec{s_0}$ and $\vec{q}$. The annotation $\tilde{z}$ of $z$ is obtained by concatenating each of these arrays. In practice, to avoid the transformer block (context) size from growing too large, we fix a cutoff $T$ and first trim each array to its first $T$ elements.

We formalize this in the terminology of Appendix D by defining a Transcript Annotator and Answer Extractor. Note that, since our proof system consists only of one "answer" $z$ send from the prover to the verifier, the entire transcript $\pi$ is simply $z = (z_0, z_1)$. Since the verification is deterministic, this means that the proof system is of an NP type (however, note that the search problem of finding the "NP-witness" $z = (z_0, z_1)$ is in fact in P).

- *Transcript Annotator A:* For a fixed cutoff $T$ and given input $x = (x_0, x_1)$ and transcript $z = (z_0, z_1)$, $A$ executes Algorithm 4 on input $x = (x_0, x_1)$. During the execution, $A$ stores the first $T$ intermediate values of $r_0$, $s_0$ and $q$ in arrays $\vec{r_0}$, $\vec{s_0}$ and $\vec{q}$. It outputs $A(x, z) := (\vec{r_0}, \vec{s_0}, \vec{q}, z)$.

- *Answer Extractor E:* Given an annotated transcript $\tilde{z} = (\vec{r_0}, \vec{s_0}, \vec{q}, z)$, outputs $E(\tilde{z}) := z$.

We note that the computational complexity of $A$ is roughly that of the honest prover, i.e., Algorithm 4 (up to additional space due to storing intermediate values). As for $E$, it can be implemented in logarithmic space and linear running time in $|\tilde{z}|$, i.e., the length of the description.[14]

## F  EXPERIMENT DETAILS

We provide details of how we implemented the experiments in Section 5 and additional figures for each experiment. Code, data and models are available as supplementary material.

**Model architecture.** We use Karpathy's *nanoGPT*[15] implementation of GPT. Note that we train the model "from scratch" only on sequences related to the GCD problem, rather than starting from a pretrained checkpoint. We use a 6.3M parameter architecture of 8 layers, 8 attention heads, and 256 embedding dimensions. We optimized hyperparameters via a random hyperparameter search, arriving at learning rate 0.0007, AdamW $\beta_1 = 0.733$ and $\beta2 = 0.95$, 10% learning rate decay factor, no dropout, gradient clipping at 2.0, no warmup iterations, and 10% weight decay.

**Data.** We sample integers from the $\log_{10}$-uniform distribution over $\{1, \ldots, 10^4\}$. Models in Table 2 and Fig. 2 are trained for 100K iterations on a dataset of $\approx$10M samples. For Figure 3 (base ablation) we train for 20K iterations on a dataset of $\approx$1M samples; this is because this setting required 68 many runs in total, whereas the annotation-cutoff ablation required 18 longer runs.

**Compute.** All experiments were run on a machine with an NVIDIA A10G GPU, 64GB of RAM, and 32 CPU cores. The longest experiment was the single RLVF run, which took one month and

---

[14]That is, if integers are represented by $n$-bits, then $E$ has space complexity $O(\log n + \log T)$ and running time $O(n \cdot T)$.

[15]https://github.com/karpathy/nanoGPT.

four days. The annotation-cutoff ablation runs took about 75 minutes each. Base of representation ablation runs were shorter at about 15 minutes each. The total running time of the Transcript Learning experiments was approximately 40 hours (excluding time dedicated to a random hyperparameter search), and the RLVF experiment took another month and four days. The overall disk space needed for our models and data is 4GB.

**Representing integers.** We fully describe how integer sequences are encoded. As a running example, we will use base 210. To encode a sequence of integers, each integer is encoded in base 210, a sign is prepended and a delimiter is appended, with a unique delimiter identifying each component of the sequence. For example, consider the input integers $x_0 = 212$ (which is 12 in base 210) and $x_1 = 159$. Their GCD is $y = 53$, with Bézout coefficients $z_0 = 1$ and $z_1 = -1$. Therefore, the sequence $(212, 159, 53, 1, -1)$ is encoded as

$$+,1,2,\texttt{x0},+,159,\texttt{x1},+,53,\texttt{y},+,1,\texttt{z0},-,1,\texttt{z1}$$

where commas are added to distinguish between different tokens. Null tokens are appended to pad all sequences in a dataset to the same length. Both the input and the padding components are ignored when computing the loss and updating parameters.

**Annotations** Annotations are encoded as above, with each component in an intermediate step $\pi_t$ delimited by a unique token. Since different integer pairs may require a different number of intermediate steps to compute the Bézout coefficients, we chose to pad all annotations to the same length $T$ by the last step $\pi_T$ in the sequence (which consists of the final Bézout coefficients). This ensures that the final component output by the model in each sequence should be the Bézout coefficient, and allows us to batch model testing (generation and evaluation) resulting in a 1000x speed-up over sequential testing.

As an example, consider the inputs $x_0 = 46$ and $x_1 = 39$. Tracing through the execution of Algorithm 4, we have

| $x_0$ | $x_1$ | $y$ | $\vec{s_0}$ | $\vec{r_0}$ | $\vec{q}$ | $z_0$ | $z_1$ |
|-------|-------|-----|-------------|-------------|-----------|-------|-------|
| 46 | 39 | | 1 | 46 | 1 | | |
| | | | 0 | 39 | 5 | | |
| | | | 1 | 7 | 1 | | |
| | | | $-5$ | 4 | 1 | | |
| | | | 6 | 3 | 3 | | |
| | | 1 | | | | $-11$ | 13 |

To encode this as an annotated transcript for the transformer, we must specify a base of representation and an annotation cutoff. Suppose that we wish to encode this instance in base $B = 10$ and cutoff $T = 3$. Then the input with the annotated transcript is encoded as

$$\begin{aligned}
&+,4,6,\texttt{x0},+,3,9,\texttt{x1},+,1,\texttt{y},\\
&+,1,\texttt{z0'},+,4,6,\texttt{z1'},+,1,\texttt{q'},\\
&+,0,\texttt{z0''},+,3,9,\texttt{z1''},+,5,\texttt{q''},\\
&+,1,\texttt{z0'''},+,7,\texttt{z1'''},+,1,\texttt{q'''},\\
&-,1,1,\texttt{z0},+,1,3,\texttt{z1}
\end{aligned}$$

where commas are used to separate between tokens, and linebreaks are added only for clarity. Notice the three types of tokens: signs, digits, and delimiters. Notice also that the output $y$ is added immediately after the input, followed by the annotated transcript (whose six tokens comprise the proof itself). Since the Self-Proving model we train has causal attention masking, placing the output $y$ before the proof means that the model "commits" to an output and only then proves it.

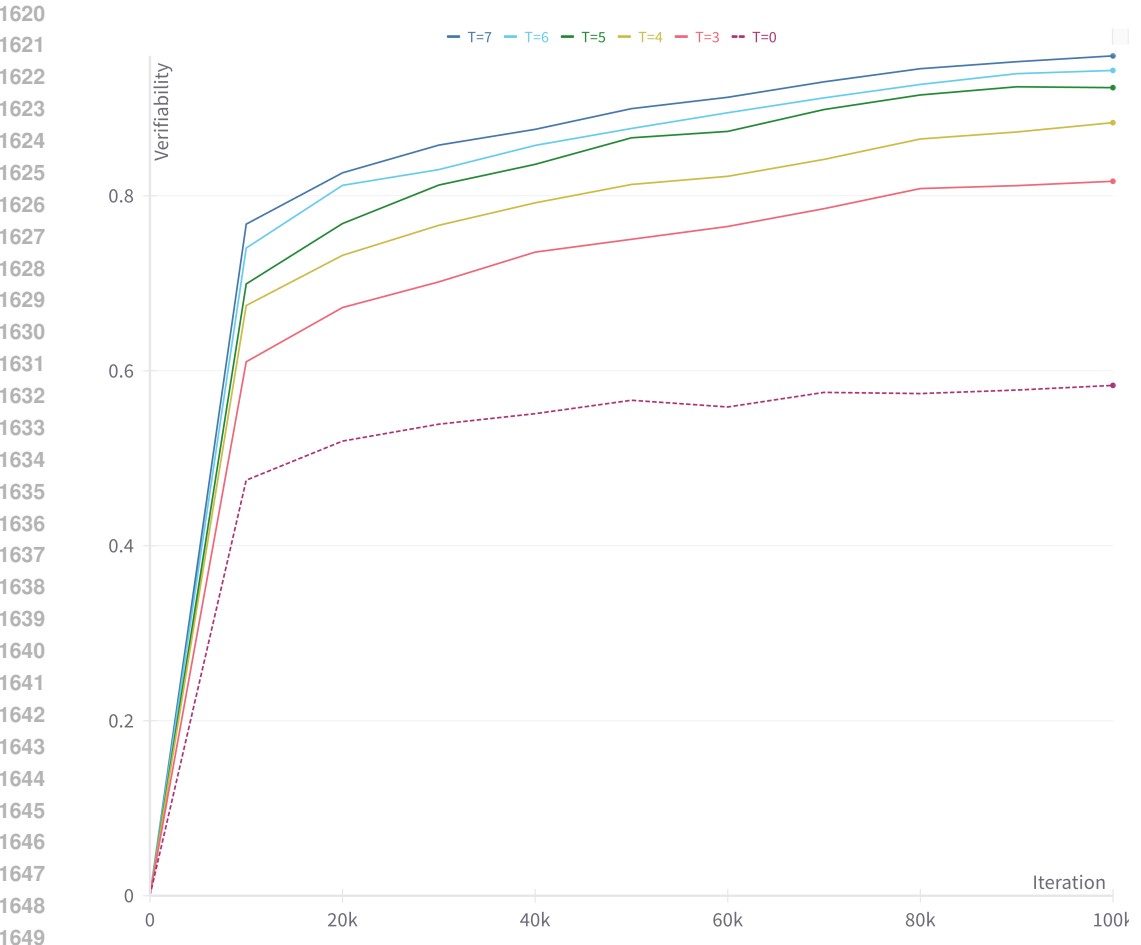

Figure 5: **Verifiability as a function of the number of samples** $N$. Each iteration (X axis) is a batch of 1024 samples from a dataset of $\approx$10M sequences. Every 10k iterations, Verifiability was evaluated on a held-out dataset of 1k inputs (as described in Section 5). $T$ is the number of steps in Annotated Transcript Learning (Figure 2), and $T = 0$ is non-annotated Transcript Learning. Each $T$ was run with three seeds, with mean depicted by the curve and standard error by the shaded area.

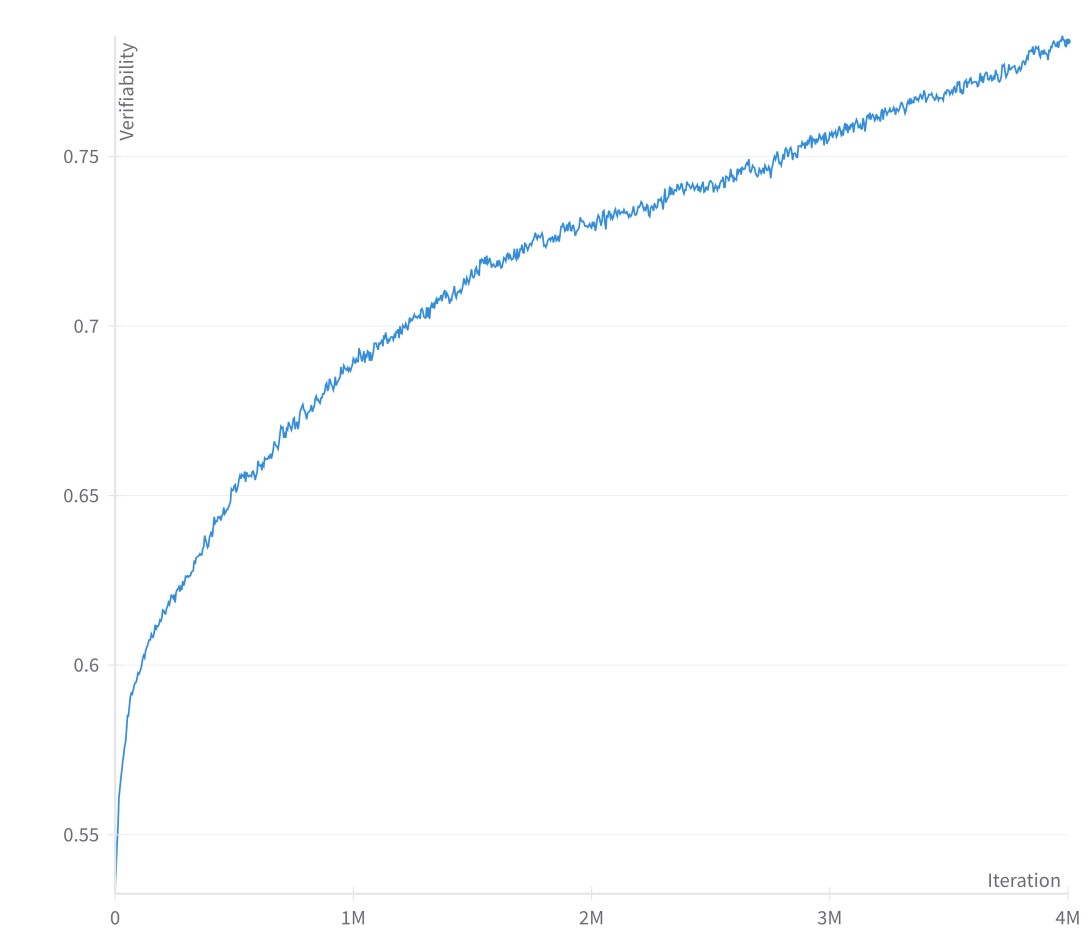

Figure 6: **RLVF Verifiability as a function of the number of samples** $N$. Starting from a base model with Verifiability 48% (obtained via Transcript Learning), in each iteration a batch of 2048 inputs are sampled; the model generates a proof for each; the Verifier is used to check which proofs are accepted; then, the model parameters are updated accordingly (see Algorithm 2). Verifiability was evaluated on a held-out dataset of 1k inputs.

