# OpenReview forum: "Models That Prove Their Own Correctness"
_ICLR.cc/2025/Conference — Submitted to ICLR 2025_

### Official Review · Reviewer_jLku · 2024-10-31

**Soundness:** 3
**Presentation:** 3
**Contribution:** 3
**Rating:** 6
**Confidence:** 4

**Summary:**

In this work, the authors propose a methodology to learn models that can provide evidence about the correctness of their answers.
Given a function F:Sigma*->Sigma*, a model for F is a function F_theta that assigns to each x\in Sigma^* a probability distribution F_theta(x).

A model is alpha-correct if on a random input x, the output F_{theat}(x) is equal to F(x) with probability at least \alpha.
Given a function F and a verifier V, a model F_theta is beta-self proving if V(x,y)=1 with hight probability, where x is an input sampled at random and y is sampled at random from F_theta(x).

The goal is to learn a model that has a high degree of correctness (\alpha close to 1) and a high degree of verifiability (\beta close to 1). The authors develop a methodology for this task, and use learning models that compute the GCD of two numbers as an example.

**Strengths:**

I find the topic of the paper quite interesting.

 The paper seems to provide a nice framework to combine interactive proof systems with techniques from learning theory. Although formal verification has been widely studied in connection with learning theory, the disadvantage is that, in most of the approaches, the goal is to construct a proof of correctness in some fixed proof system. The use of interactive proof systems adds a lot of flexibility to the process, besides giving an avenue for a theoretical analysis related to the convergence of the learning process etc.

**Weaknesses:**

The disadvantage of the approach is that it requires access to the implementation of a previously existing verifier. I find that the paper lacks a discussion about the usefulness of trying to learn a function for which we already have an implementation.

**Questions:**

No questions.

---

> ### Author Response · Authors · 2024-11-18
>
> Dear Reviewer,
>
> Thank you for highlighting the flexibility of our framework, which captures any Interactive Proof system. Indeed, as you note, one of the main contributions of our paper is creating an avenue for theoretical analysis and provable guarantees.
>
> Regarding the weakness you raised:
> > The disadvantage of the approach is that it requires access to the implementation of a previously existing verifier. I find that the paper lacks a discussion about the usefulness of trying to learn a function for which we already have an implementation.
>
> We would like to clarify an important distinction: A model with high correctness ($\alpha$ close to 1) may not necessarily be Self-Proving with respect to a given verifier $V$. In fact, if the model was trained without any concern for $V$, we cannot expect it to be verifiable by $V$. The main technical contribution of our paper is proposing and analyzing (theoretically and empirically) two methods for fitting the model to the verifier.
>
> Crucially, the task of Self-Proving models is not to "learn a function," but rather to learn how to generate proofs that the function was computed correctly. This distinction is fundamental: The usefulness of Self-Proving models lies in their ability to not only generate an output (compute the function), but also prove that the output was correct.
>
> We discuss this distinction through an example in the paper, at the bottom of page 5 (immediately before Section 4). Does this address your concern?

---

> ### Author Response · Authors · 2024-11-26
> **Follow-up on rebuttal: Reviewer jLku**
>
> Dear Reviewer jLku,
>
> As we are approaching the revision deadline (November 27th), we wanted to follow up on our rebuttal from November 18th. Our rebuttal and revision resolve the concerns that you raised. We would appreciate if you could acknowledge our rebuttal and, if you agree, consider updating your score.
>
> Best regards,
> The authors

---

### Official Review · Reviewer_TyWJ · 2024-11-03

**Soundness:** 2
**Presentation:** 2
**Contribution:** 2
**Rating:** 3
**Confidence:** 3

**Summary:**

This paper proposes a self-proving model, a new learning paradigm that a model outputs both a prediction $y$ and a proof for the correctness of the prediction by interacting with a verifier. A self-proving model is useful for users since it can ensure that the output of a model is correct. The paper proposes a self-proving model and defines some metrics for evaluating the performance of a self-proving model. The paper also gives learning algorithms for learning a self-proving model and theoretical analyses on the convergence of the transcript learning algorithm. The experiments evaluate the performance of self-proving models on a GCD task.

**Strengths:**

1. This is a well-motivated work. The use case shown in the introduction of the paper is attractive.

2. The proposed self-proving framework that proves the correctness of the answer by interactions between an autoregressive model and an external verifier seems a natural setting.

**Weaknesses:**

1. Theorem 4.1, one of the major theoretical contributions of the paper, makes some strong assumptions. Therefore, I think the contribution of the theorem is limited.  Firstly, it assumes that $A(\theta)$ is concave. I think this assumption does not hold for the typical autoregressive models used today, including the GPT model used in experiments. Moreover, the theorem assumes the existence of $\theta^\ast$ satisfying $A(\theta^\ast) \geq 1 - \epsilon/2$. This is also a strong assumption since it is currently not clear whether such self-proving models exist or not.
2. The paper evaluates the self-proving model's performance on a GCD task. However, I feel that GCD would not be appropriate as a use case for a self-proving model since it is an easy task, and we do not need any machine-learning techniques to solve it. It is reasonable that Carton (2024) solved a GCD task since the paper's main objective is to understand how a transformer works. On the other hand, I think that this paper should show the effectiveness of the proposed self-proving model, and the experiments with a GCD task are not sufficient.
3. The paper emphasizes the use-case that self-proving models can guarantee the correctness of a specific $x_0, y_0$. (line 262). This feature depends on  the $s$-soundness defined for a probabilistic verifier. However, the assumption that $s$-soundness holds for any $x, y, P$ is unrealistic (See the question 2 below). It is more natural to assume that a false-positive error (line 207) depends on the distribution over $x, y$. However, if we make such an assumption, then it is difficult to give a guarantee for specific $x_0, y_0$. Therefore, I think the self-proving model does not work as stated in the use case.

**Questions:**

1. I think this type of theoretical bound needs a confidence parameter $\delta$. Since we estimate parameter $\theta$ from a finite set of $N$ samples instead of accessing the distribution $\mu$,  it is possible that the drawn samples are "bad" and that we cannot estimate suitable parameters from the samples. Could you please explain why we do not need a confidence parameter $\delta$?
2. The paper says that completeness and soundness are properties of a verifier (Definition 3.2). However, it seems unrealistic to imagine a probabilistic verifier whose soundness error is always smaller than $s$ for *any* $x, y$ and $P$, unless $s = 0$. How can we obtain such a verifier? Where does the probability come from?  Moreover, I think it is possible to reduce false positive errors (line 208) arbitrarily by running a probabilistic verifier $V$ for multiple times for the same $(x, y, P)$. It is more realistic for me to assume that there are specific $(x, y, P)$ that causes a false-positive error for the verifier $V$, and thus it depends on the distribution $\mu$.
3. Related to the above point, Definition 3.2 assumes the randomness of $V$ and $P$ line 209, but the definition of soundness assumes the condition holds for all $P$. Is $P$ random?

---

> ### Author Response · Authors · 2024-11-18
> **Response (part 1)**
>
> Thank you for your thoughtful review. Your review raises important questions about three aspects of our work: (1) the theoretical assumptions in Theorem 4.1, particularly regarding concavity and existence of high-agreement models, (2) the scope of our experimental evaluation, and (3) the practicality of worst-case soundness guarantees. We welcome the opportunity to address these concerns and show how our theorem, while built on strong assumptions, provides valuable insights that are supported by empirical evidence. We also explain how worst-case guarantees, far from being unrealistic, are both achievable and crucial for reliable verification. Below, we address each point in detail, along with your specific questions about confidence parameters and verifier properties.
>
> ### 1. Concavity of $A(\theta)$
> > Firstly, it assumes that $A(\theta)$ is concave. I think this assumption does not hold for the typical autoregressive models used today, including the GPT model used in experiments.
>
> You are correct that deep neural networks, including GPT-style autoregressive models, generally have non-convex loss landscapes. However, we believe that there is still value of analyzing convex cases, even when studying non-convex systems:
>
> 1. Theoretical foundations: Convex analysis provides clean mathematical tools that help build intuition about the fundamental properties and limits of these learning problems, even when the exact assumptions don't hold in practice. Many seminal works in machine learning (e.g., early theory of SVMs) started with convex analyses that later inspired broader non-convex results.
> 2. Stepping stone: This work serves as a first step toward understanding the more general non-convex case. Similar to how convex optimization theory informed the development of non-convex optimization methods, our convex analysis could highlight important properties that generalize or inspire future non-convex analyses.
>
> We remark that these points seem to be shared by other theoreticians in our field. These concerns are usually addressed by the inclusion of experiments to theoretical papers, as we have done in our paper. Indeed, our experiments demonstrate convergence of TL to a Self-Proving model, despite the non-convex (or rather, non-concave) optimization landscape. Following your suggestion, we will add a generalization of our theory to non-convex settings (e.g. as in [1,2,3]) in our discussion of future work.
>
> ### 2. Existence of high-agreement $\theta^\ast$
> > Moreover, the theorem assumes the existence of $\theta^\ast$ satisfying $A(\theta^\ast) \geq 1 - \eps/2$. This is also a strong assumption since it is currently not clear whether such self-proving models exist or not.
>
> This assumption says that there is an instantiation of the model parameters (e.g., weights and biases of the neural network) that result in a $(1-\eps/2)$ Self-Proving model. We observe that is an almost necessary assumption, in the following sense. Let us take $\eps = 2%$ and think about transformers models, for simplicity: the goal is to train the transformer to be $98%$ Self-Proving. If the assumption does *not* hold, it means that there is *no* weights and biases which are $99%$ Self-Proving. That is, even if we were to exhaustively search over all parameters, we would never end up with error less than $1%$. In this case, there is (formally) no hope for learning a Self-Proving model; instead, we should try a different architecture altogether.
>
> We hope this explanation fully addresses your concern, and welcome any follow-up questions.
>
> ### 3. Additional experiments
> > I feel that GCD would not be appropriate as a use case for a self-proving model since it is an easy task, and we do not need any machine-learning techniques to solve it. It is reasonable that Carton (2024) solved a GCD task since the paper's main objective is to understand how a transformer works. On the other hand, I think that this paper should show the effectiveness of the proposed self-proving model, and the experiments with a GCD task are not sufficient.
>
> We will add an additional experiment on a significantly more challenging problem, namely, Quadratic Residuosity. Since this request was shared by other reviewers, we describe this experiment in the "Overall Comment." We welcome any feedback on the proposed experiment. Due to computational constraints, we may not by able to conclude the experiments by the end of the discussion period, but we will include the results in the camera ready version of the paper.

---

> > ### Author Response · Authors · 2024-11-18
> > **Response (part 2)**
> >
> > ### 4. On worst-case soundness guarantees
> > > the assumption that $s$-soundness holds for any $x, y, P$ is unrealistic (See the question 2 below). It is more natural to assume that a false-positive error (line 207) depends on the distribution over $x,y$. However, if we make such an assumption, then it is difficult to give a guarantee for specific $x_0, y_0$. Therefore, I think the self-proving model does not work as stated in the use case... The paper says that completeness and soundness are properties of a verifier (Definition 3.2). However, it seems unrealistic to imagine a probabilistic verifier whose soundness error is always smaller than  $s$ for any $x,y$ and $P$, unless $s=0$. ... It is more realistic for me to assume that there are specific $(x,y,P)$ that causes a false-positive error for the verifier, and thus it depends on the distribution $\mu$.
> >
> > The strength of Interactive Proof systems (IPs) lies precisely in their worst-case soundness guarantees---the very property you reference. While average-case guarantees based on benchmark performance are valuable, worst-case guarantees protect users from incorrect outputs regardless of the input distribution. Importantly:
> >
> > 1. Worst-case guarantees are strictly stronger than average-case guarantees, ensuring Self-Proving models function as intended in all use cases.
> > 2. Such guarantees are realistic for an extremely broad class of problems: any computation feasible in polynomial space admits a probabilistic proof system [4,5].
> >
> > Regarding specific questions:
> >
> > >  I think it is possible to reduce false positive errors (line 208) arbitrarily by running a probabilistic verifier $V$ for multiple times for the same $(x,y,P)$.
> >
> > Your observation is correct: soundness error can be made exponentially small through independent repetition of the verification procedure.
> >
> > > How can we obtain such a verifier?
> >
> > There is a rich literature, spanning cryptography and complexity theory, on designing efficient verifiers for different notions of "efficiency" and different computational problems. Efficient verifiers are known both for many specific problems with algebraic or combinatorial problems, and for general classes of computations. This literature has put forward many ideas and tools that can be used to derive new verifiers for problems (and efficiency measures) of interest. Thus, deriving a new verifier could proceed either by expert design, or via automation by using results that apply to classes of computations. We also refer the reader to a primer on probabilistic proof systems [4], which presents specific proof systems and their verifiers, as well as the general power and limits of probabilistic verification.
> >
> > > Where does the probability come from?
> >
> > The probability in the definition of soundness comes from the randomness used by the Verifier: We consider probabilistic Verifiers, whose questions are randomly generated during interaction with the prover. A concrete example of probabilistic verification can be found in the proof system presented in our Overall Comment. Efficient verification of certain complex problems *requires* randomness, i.e., nonzero soundness error. As you observed, the soundness error can be made exponentially small by repeating the verification. We are more than happy to provide further explanation upon request.
> >
> > > Definition 3.2 assumes the randomness of $V$ and $P$ line 209, but line 209, but the definition of soundness assumes the condition holds for all $P$. Is $P$ random?
> >
> > We define soundness against deterministic provers without loss of generality. Since soundness is guaranteed against computationally unbounded provers, any randomized cheating prover can be converted to a deterministic one that simply uses the optimal random choices.
> >
> > ### 5. Confidence of parameter for theorem
> > > I think this type of theoretical bound needs a confidence parameter $\delta$. Since we estimate parameter $\theta$ from a finite set of $N$ samples instead of accessing the distribution $\mu$, it is possible that the drawn samples are "bad" and that we cannot estimate suitable parameters from the samples. Could you please explain why we do not need a confidence parameter
> >
> > This is because we show convergence "in expectation" rather than "with high probability." This follows directly from our reduction to SGD in the convex setting---the classical SGD convergence theorems we use (which require boundedness, Lipschitzness, and convexity) are themselves stated in expectation rather than with high probability.
> > An interesting direction for future work would be to derive high-probability bounds, which would indeed require a confidence parameter $\delta$. This would likely require different proof techniques and potentially stronger assumptions, but could provide "with high probability" convergence guarantees.

---

> > > ### Author Response · Authors · 2024-11-18
> > > **Response (part 3)**
> > >
> > > ### References
> > >
> > > 1. Gradient Descent Finds Global Minima of Deep Neural Networks. Simon Du, Jason Lee, Haochuan Li, Liwei Wang, Xiyu Zhai. ICML 2019.
> > > 2. Convexity, Classification, and Risk Bounds. Peter L. Bartlett, Michael I. Jordan, Jon D McAuliffe. Journal of the American Statistical Association 2012.
> > > 3. Better Theory for SGD in the Nonconvex World. Ahmed Khaled, Peter Richtárik. TMLR 2023.
> > > 4. IP=PSPACE. Adi Shamir. J. ACM 1992.
> > > 5. Algebraic Methods for Interactive Proof Systems. Carsten Lund, Lance Fortnow, Howard J. Karloff, Noam Nisan. J. ACM 1992.

---

> > > > ### Comment · Reviewer_TyWJ · 2024-11-23
> > > >
> > > > Thank you for addressing my concerns. Some of my concerns were solved, but I will keep my score unchanged for the following reasons:
> > > >
> > > > 1. Theorem 4.1 makes strong assumptions and has a gap with practical settings. Hence, they seem less important.
> > > > 2. The motivation for solving GCD and Msqrt is unclear. If we want to solve these problems, there is a better way that does not use ML.
> > > > 3. The experimental results are limited to a single simple task, so they seem weak in supporting the effectiveness of the proposed approach. The results of Msqrt have not yet been reported, and we currently have no evidence that the experiments will work well.
> > > >
> > > >
> > > > ## Concerns on premises of Theorem 4.1
> > > >
> > > > I agree that assuming concavity is a cornerstone for many problems. However, we also have to consider the gap size between a simplified setting and the real one. I believe assuming concavity on non-autoregressive LLMs like GPTs is unrealistic. There is a large gap, and the theoretical results are less meaningful accordingly.  Moreover, experimental evaluations are limited to a simple setting and are also not exhaustive enough to validate the theoretical findings.
> > > >
> > > > The authors say that the assumptions on $\theta^\ast$ are necessary to draw a conclusion. But since both self-proving and transcript learning are new concepts, it is unclear when assuming the existence of such $\theta^\ast$ is reasonable. Clearly, there are practical problems that are hard or impossible to verify. The paper should show the conditions that assuming the existence of $\mathcal{T}^\ast$ and $\theta^\ast$ is reasonable.
> > > >
> > > >
> > > > What theorem 4.1 says is: "If transcript learning is possible with some parameter $\theta^\ast$ and the objective is concave, then we can estimate a good parameter $\bar{\theta}$.". I agree that this theorem is valuable if we agree with the premises. However, (1) the objective is not concave generally, and (2) whether self-proving or $\mathcal{T}^\ast$ are possible in general or not is unclear. Therefore, I think the assumptions made on the theorem are strong, and the theoretical results are less important.
> > > >
> > > >
> > > >
> > > > ## Concerns on experiments.
> > > >
> > > > Experimental evaluation does not show an appropriate use case for a self-proving system. GCD is a problem that we generally do not use machine learning to solve since there are strong non-ML methods for solving it. Moreover, experimental evaluations are limited to a single task. The additional problem setting Msqrt provided by the authors is interesting, but it lacks experimental evaluation results. Moreover, I think the additional task is also not well-motivated since there exist strong non-ML methods for solving it.
> > > >
> > > >
> > > > ## Others
> > > > - I understand the concept of a probabilistic verifier. I agree with the authors that considering s-soundness is important for efficiency.
> > > > - I understand that we do not need a confidence parameter since we discuss expected values.

---

> > > > > ### Author Response · Authors · 2024-11-26
> > > > >
> > > > > We appreciate your thoughtful feedback and are encouraged that we've resolved most of your concerns. Let us address the remaining two concerns directly.
> > > > >
> > > > > # Assumptions of Theorem 4.1
> > > > > We follow your decoupling of your concern into (1) convexity of optimization landscape, and (2) the realizability assumption.
> > > > >
> > > > > ### Convexity
> > > > > We agree that LLM training is, in general, a nonconvex problem, and we have modified the text to make it clear that our theorem does not always match ML tasks which emerge in reality. We note, however, that convexity is a widely-used assumption for proving gradient-descent convergence. We are not aware of (nonconvex) LM convergence theorems in the literature, though we welcome any pointers if they exist.  Convergence without convexity is the subject of cutting-edge papers [1,2,3]---which dedicate their entirety to the issue of nonconvexity.
> > > > >
> > > > > Back to our case, the convexity-based theory helps us demonstrate the viability of Transcript Learning, which we then empirically validate in the nonconvex setting. In other words, the theorem helps us understand Transcript Learning in a simplified setting which can be mathematically analyzed.
> > > > >
> > > > > ### Realizability
> > > > > The realizability assumption is a necessary constraint: if a Self-Proving model cannot be realized within the chosen architecture, then learning such a model is impossible regardless of the training approach. Rather than being a limitation that requires justification, it represents a necessary logical precondition.
> > > > >
> > > > > The concern then lies in selecting an architecture capable of expressing a Prover for a given Proof System. One common approach assumes deep neural networks as universal function approximators, scaling both architecture size and training data until achieving desired performance. Recent theoretical work has established rigorous foundations for this approach, demonstrating the Turing-completeness of transformers [6] and their variants [7]. These architectures can even approximate arbitrary continuous sequence-to-sequence functions on compact domains [8]. Therefore, transformer architectures can realize any Turing machine---including the Prover in an Interactive Proof system, which is a polynomial-space Turing machine (or better [9]).
> > > > >
> > > > > We have added this discussion to the paper---thank you for prompting this valuable clarification.
> > > > >
> > > > > # Experiments
> > > > > Our small-scale experiments are a proof of concept, validating our theorem that shows that proofs can in fact be learned from accepting transcripts. In particular, a growing body of research is concerned with the ability of LLMs to learn arithmetic problems. Several works studied the possibility of solving arithmetic problems, and our paper extends towards proving solutions.
> > > > >
> > > > > Indeed, as in other arithmetic problems, GCD has efficient classical algorithms; our goal is not to outperform existing methods but rather to demonstrate that neural networks can learn to both solve and prove correctness of their solutions. This capability is crucial as we scale to more complex problems where classical algorithms may not exist or may be intractable. GCD serves as an ideal test case because we can verify our results against ground truth while developing the fundamental techniques needed for harder problems.
> > > > >
> > > > > Regarding the upcoming MSqrt experiments: MSqrt is particularly relevant precisely because it lacks efficient worst-case algorithms (it is believed and widely assumed to be intractable in the non-quantum setting) when $N$ is composite. The case of prime $N$ is also relevant, since the cost of finding a square root of $x$ is still higher than verifying that y is a square root.
> > > > > When you mention 'strong non-ML methods' for MSqrt, are you perhaps referring to algorithms that work well in practice? We'd be very interested in understanding which specific strong methods you had in mind.
> > > > >
> > > > > Finally, regarding the following comment:
> > > > > > The results of Msqrt have not yet been reported, and we currently have no evidence that the experiments will work well.
> > > > >
> > > > > Proposing MSqrt experiments in response to reviewers concerns, and before knowing their outcome, reflects good scientific practice. The results--—whether positive or negative--—will provide valuable insights about our theory's practical viability, particularly given MSqrt's step-up in complexity over GCD.
> > > > >
> > > > > # References
> > > > > 6. On the Computational Power of Transformers and Its Implications in Sequence Modeling. Satwik Bhattamishra, Arkil Patel, Navin Goyal. CoNLL 2020.
> > > > > 7. Universal Transformers. Mostafa Dehghani, Stephan Gouws, Oriol Vinyals, Jakob Uszkoreit, Łukasz Kaiser. ICLR 2019.
> > > > > 8. Are Transformers universal approximators of sequence-to-sequence functions? Chulhee Yun, Srinadh Bhojanapalli, Ankit Singh Rawat, Sashank J. Reddi, Sanjiv Kumar. ICLR 2020.
> > > > > 9. Delegating Computation: Interactive Proofs for Muggles. Shafi Goldwasser, Yael Tauman Kalai, Guy N. Rothblum. J. ACM 2015.

---

> > > > > > ### Comment · Reviewer_TyWJ · 2024-11-26
> > > > > >
> > > > > > Thank you for your further response.
> > > > > >
> > > > > > > Convexity
> > > > > >
> > > > > > My opinion is unchanged for this part: assuming convexity on LLMs like GPTs seems unrealistic. Therefore, the theory is less important.
> > > > > >
> > > > > > > Realizability
> > > > > >
> > > > > > If we acknowledge the realizability assumption, it means there always exists $\theta^\ast$ that leads to a correct proof. I think this is strange since not all the predictions a model makes can be proved. As I wrote in the previous comment, the paper should show the conditions that assuming the existence of $\mathcal{T}^\ast$ and $\theta^\ast$ is reasonable.
> > > > > >
> > > > > > > Experiments
> > > > > >
> > > > > > I'm sorry for the misunderstanding. I agree that there is no strong algorithm for the MSqrt problem. However, this will not change my overall opinion of the experiments: Experiments on GCD are not well-motivated,  and evaluations are insufficient to show that the proposed self-proving works well in situations where we want it.
> > > > > >
> > > > > > > The results--—whether positive or negative--—will provide valuable insights about our theory's practical viability,
> > > > > >
> > > > > > I agree that both positive and negative results are important for research progress. On the other hand, if the proposed method does not work well with difficult problems, then the proposed self-proving model becomes less attractive. Therefore, the experimental results are important for assessing the paper's impact. It might be interesting to understand why it fails, but it would require further analyses and another round of peer review.

---

> > > > > > > ### Author Response · Authors · 2024-11-28
> > > > > > >
> > > > > > > We thank the reviewer for continuing this discussion. Through our exchanges, we resolved the majority of the original concerns, including questions on the viability of worst-case soundness guarantees, probabilistic verification, the lack of a confidence parameter $\delta$, and the computational hardness of MSqrt.
> > > > > > >
> > > > > > > We address the three remaining concerns below and will happily provide further clarification. However, at this point we would like to note that the remaining concerns stem from broader questions about the role of theoretical analysis in our field, and different interpretations of ML theory notions (namely, realizability). While these points merit discussion, they may be difficult to reconcile through a continued technical discussion alone.
> > > > > > >
> > > > > > > ### Convexity
> > > > > > > > My opinion is unchanged for this part: assuming convexity on LLMs like GPTs seems unrealistic. Therefore, the theory is less important.
> > > > > > >
> > > > > > > This critique would apply equally to any of the many papers in ML theory that use convexity to obtain convergence guarantees. Our contribution follows the widely-accepted and well-established tradition of analyzing simplified settings---which we have then empirically validated.
> > > > > > > In sum, the concern surrounding the simplifying convexity assumption appears to stem from a fundamental difference in how we view the role and value of theoretical (computer) science, where simplifying assumptions are standard practice for establishing foundational results.
> > > > > > >
> > > > > > > ### Realizability
> > > > > > > > If we acknowledge the realizability assumption, it means there always exists $\theta^\ast$ that leads to the correct proof.  I think this is strange since not all the predictions a model makes can be proved.
> > > > > > >
> > > > > > > There seems to be a misunderstanding about the nature of realizability assumptions. The purpose is not to assert that *any* model family contains a perfect model. Rather, it states that the convergence bound holds for model families that are sufficiently expressive. As we explained in the previous response, this is a logical necessity.
> > > > > > >
> > > > > > > > As I wrote in the previous comment, the paper should show the conditions that assuming the existence of $\mathcal{T}^\ast$ and $\theta^\ast$ is reasonable
> > > > > > >
> > > > > > > As explained in our previous response and incorporated in the revision, recent results establishing the Turing-completeness of transformers provide theoretical justification: any Honest Prover can be realized by a transformer. Moreover, as explained in the paper (lines 310-316), an honest transcript generator $\mathcal{T}^\ast$ can be realized e.g. in the case of Doubly-efficient Interactive Proof systems.
> > > > > > >
> > > > > > > ### Experiments
> > > > > > > > I'm sorry for the misunderstanding. I agree that there is no strong algorithm for the MSqrt problem.
> > > > > > >
> > > > > > > We appreciate this acknowledgement.
> > > > > > >
> > > > > > > > However, this will not change my overall opinion of the experiments: Experiments on GCD are not well-motivated, and evaluations are insufficient to show that the proposed self-proving works well in situations where we want it.
> > > > > > >
> > > > > > > We maintain that GCD serves as a valuable proof-of-concept motivated by a rich literature on the arithmetic capabilities of transformers, with MSqrt providing a natural next step toward more challenging problems.

---

> > > > > > > > ### Comment · Reviewer_TyWJ · 2024-12-01
> > > > > > > >
> > > > > > > > Thank you for your further response.
> > > > > > > >
> > > > > > > > Yes, the remaining concerns are the experiments and theoretical results. I think both are weak and thus, I feel it is difficult to recommend this paper for acceptance.
> > > > > > > >
> > > > > > > > I agree there are cases assuming convexity is reasonable. However, I think this is not the case since the paper assumes the use of LLMs. They are not convex and, in my opinion, far from it. Since the proposed method assumes the use of LLMs, assuming convexity is unrealistic. Please note that I'm not saying that the theoretical results are meaningless, but they are weak.
> > > > > > > >
> > > > > > > > > Assumptions on $\theta$ and $\mathcal{T}^\ast$
> > > > > > > > The authors said that any prover and transcript generator are possible. However, as I wrote in the previous response, I strongly believe that self-proving cannot be applied to all situations where we use an ML model to make a prediction $y$ from input $x$. For example, I think it is hard to give a proof to a machine translation task.
> > > > > > > >
> > > > > > > > Self-proving is an interesting concept, but the paper must show when we can use self-proving. Currently, the limitation is unclear, and the only case the paper shows is GCD. Therefore, it is hard for me to judge its usefulness.

---

### Official Review · Reviewer_W6ru · 2024-11-03

**Soundness:** 3
**Presentation:** 3
**Contribution:** 3
**Rating:** 6
**Confidence:** 3

**Summary:**

The paper develops the concept of self-proving models that justify (or "prove" in the authors language) their answers to a trusted and pre-built verifier. The process results in a probablistic guarantee on the correctness of the answer provided by the self-proving model.  Transcript learning and RL are proposed as methods to derive a self-proving model. Experiments are conducted on a Greatest Common Divisor problem.

**Strengths:**

I find the overall approach principled and welcome. Moving the assessment of the answer of a model from test results to an estimate of the correctness on the individual query (and this not being provided by the model itself) is certainly welcome.

I am not sure the overall setup is novel as such (see below); however I believe the error bounds on the two learning approaches are. The results are well backed by theory and the paper does a good job at making this challenging subject as accessible as possible without trivialising the contribution.
Some experiments, although perhaps limited, are provided supporting the results.

**Weaknesses:**

It was not clear to me the extent to novelty of the overall setup of the pair prover/disprover. Obviously this is a well-known setup in many applications including those cited in the literature (perhaps the whole area of "Argumentation" in AI should also be included as it is not very far from here), but this particular instantiation with ML models and a formal verifier could be clarified.

The verifier obviously has a fundamental role here. I might have missed this but the implications of this were not clear to me (how can they be derived and at what costs).

I think the authors realise that their experimentation on GCD are limited even from a purely algebraic perspective. I think this is OK, but more thoughts into how this might or might not scale to more challenging problems or more general ones might be beneficial.

Fundamentally, the end result obtained by the method, if I understand the paper, is a probabilistic guarantee that the answer provided is correct (following a sequence of challenges to the verifier). In the domain explored (algebra) we tend to deal with true/false propositions.

**Questions:**

What are your views on probalistic gurantees for mathematical statement? Do you consider them useful, or is this setup a step along the way to a different application where probabilistic guarantees are more meaningful?

Can you comment on the importance and derivation of the verifier and highlight the ease or difficulty in deriving them for the problem in hand in combination with or independently of the self-proving model?

What are your thoughts on moving beyond GCD for a mathematical theory and beyond this other mathematical challenges?

Edits post review:

1. I acknowledge you might not agree with my comment of this being a "well-known set up". This was not meant as a criticism to the work. I think you are well aware that the general concept of prover/disprover set up has been long been around in logic (indeed, general philosophy before then) and theoretical computer science including in synthesis. The setup has also been used in AI Argumentation and many other areas. I understand that here the emphasis is different and so is the generality of the overall task. Note that pretty general theorem provers such as Isabelle have also been used in similar setups to aid computer proofs. In any case this was not a criticism but a request for clarification. I do not believe we fundamentally disagree here.
2. To me the role of the verifier appears to be pretty important. So I would encourage to explore this issue more. The concern I have is that a lot of the problem here has been pushed onto the (derivation of) the verifier). I agree with the authors, but only to some extent, that the work explores a somewhat different dimension. My point was and to some extent still is that whole apparatus appears to reside on the verifier but its construction is not necessarily obvious and not explored here. I accept that my suggestions can be left for further work.
3. I do appreciate the attempt to run further experiments on a different challenge, particularly given, as reported another referee, perhaps GCD is not entirely illustrative to show the advantages of the present approach.

---

> ### Author Response · Authors · 2024-11-18
> **Response (part 1)**
>
> Thank you for ackowledging the benefit of our principled approach. We invested significant effort into making our framing and results accessible, and were happy to read that you found them to be so. Next, we respond your major concerns and questions. We thank you for initiating this discussion, which has already helped us strengthen our paper and clarify its framing. We welcome any additional comments or questions.
>
> ### 1. Novelty
> > It was not clear to me the extent to novelty of the overall setup of the pair prover/disprover. Obviously this is a well-known setup in many applications including those cited in the literature (perhaps the whole area of "Argumentation" in AI should also be included as it is not very far from here), but this particular instantiation with ML models and a formal verifier could be clarified.
>
> We emphasize that our setup consists of a prover (Self-Proving model) and a verifier, not a pair of prover/disprover; the goal of the verifier is not to attempt to disprove a claim, but to certify it is correct via interaction with the prover or reject. Rejecting does not mean that the verifier concludes that the claim is false, only that the prover did not succeed in providing a sound proof of correctness.
>
> In contrast in a prover/disprover setup, to our understanding, a disprover aims to convince of the invalidity of the claim. The latter setup is more reminiscent of works on AI Safety via Debate Systems [1,2], which are distinct from our work as explained in our paper. Thank you for the pointer to the argumentation systems in AI. We will definitely add a reference to the argumentation literature.
>
> The notion of learning to prove has certainly been explored in previous works and our work adds to this landscape as discussed in the related work section: a learned prover (Self-Proving model) is trained to convince a verifier *via any Interactive Proof system* which is novel. There are no assumptions made on the Interactive Proof system, and therefore our theory is extremely general (it captures all of PSPACE [3]). We do not think that "Obviously this is a well-known setup in many applications". That said, we would be grateful if you could provide  pointers to any works that you believe we should reference and are close to our setup.
>
> ### 2. Role of the verifier
> > The verifier obviously has a fundamental role here. I might have missed this but the implications of this were not clear to me (how can they be derived and at what costs)... Can you comment on the importance and derivation of the verifier and highlight the ease or difficulty in deriving them for the problem in hand in combination with or independently of the self-proving model?
>
> Determining which problems admit an efficient verifier has been a central and foundational question in computational complexity theory. Depending on how "efficiency" is defined, designing a verifier can be either very straightforward (e.g. an NP (polynomial-time) verifier for 3SAT) or require years of breakthrough results (e.g. a PCP verifier for 3SAT, which only reads three bits in the proof).
>
> We attempt to address your concern in the "Scope" paragraph on page 2, immediately before section 2. To summarize, we clarify that your natural and fascinating question (how are verifiers derived and implement?) is beyond the scope of our paper. We also refer the reader to a primer on probabilistic proof systems [4], which presents specific proof systems and their verifiers, as well as the general power and limits of probabilistic verification.
>
> Your idea of deriving a verifier hand-in-hand with a Self-Proving model is very interesting. Our work shows how a Self-Proving model can be trained for a given verifier, but exploring the other direction sounds fascinating. At first glance, it reminds us of Prover--Verifier games [5], in which a verifier is jointly learned with the prover. As we emphasize in the Related Work section, this (interesting) setting is distinct from our work.
>
> ### 3. Additional experiments
> > I think the authors realise that their experimentation on GCD are limited even from a purely algebraic perspective. I think this is OK, but more thoughts into how this might or might not scale to more challenging problems or more general ones might be beneficial... What are your thoughts on moving beyond GCD for a mathematical theory and beyond this other mathematical challenges?
>
> We will add an additional experiment on a significantly more challenging problem, namely, Quadratic Residuosity. Since this request was shared by other reviewers, we describe this experiment in the "Overall Comment." We welcome any feedback on the proposed experiment. Due to computational constraints, we may not by able to conclude the experiments by the end of the discussion period, but we will include the results in the camera ready version of the paper.

---

> > ### Author Response · Authors · 2024-11-18
> > **Response (part 2)**
> >
> > ### 4. Why probabilistic proof systems?
> > > Fundamentally, the end result obtained by the method, if I understand the paper, is a probabilistic guarantee that the answer provided is correct (following a sequence of challenges to the verifier). In the domain explored (algebra) we tend to deal with true/false propositions... What are your views on probalistic gurantees for mathematical statement? Do you consider them useful, or is this setup a step along the way to a different application where probabilistic guarantees are more meaningful?
> >
> > You are correct, soundness holds only with high probability, but this probability can be made exponentially close to 1 and is controlled by the verifier (in other words the expected time before you will ever encounter an error is very long --1/error -- and you as a verifer control it). Therefore, it is as "useful" as a deterministic guarantee.  Incidentally, even though our paper allows probabilistic  verification, for some cases such as the GCD experiments, the verifier can extract a traditional deterministic correctness proof (which is a special case of general probabilistic interactive proofs).
> >
> > Your observation raises a fundamental point about efficient verification which we will emphasize in the final version. Complexity theory shows that probabilistic verification isn't a limitation, but rather a necessary feature for verifying complex problems or from getting extra features as follows:
> > - A non-zero soundness error (i.e., probabilistic verification) is necessary for efficiently verifying problems beyond NP (unless NP=PSPACE or at least NP=MA).
> > - In a deterministic proof system, interaction provides no additional power in the sense that any interactive proof system can be converted into a non-interactive one [6, Proposition 9.2].
> > - For Extra features such as making zero-knowledge proofs or succinct proofs, non-zero soundness is necessary as well.
> >
> > Thus, rather than viewing probabilistic guarantees as an intermediate step toward deterministic verification, complexity theory demonstrates their utility and necessity—--in particular for complex mathematical problems such as computing the permanent of 0-1 matrices [7].
> >
> > We emphasize also that, although our experiment is on an arithmetic capability, Self-Proving models can be used beyond a mathematical setting; as we prove in Theorem 4.1, Self-Proving models can be trained in any setting that admits a sound verification algorithm. Probabilistic interactive verifiers have been recently studied, e.g., in AI Safety contexts [8].
> >
> >
> > # References
> > 1. AI safety via debate. Geoffrey Irving, Paul Christiano, Dario Amodei. arXiv 2018.
> > 2. Scalable AI Safety via Doubly-Efficient Debate. Jonah Brown-Cohen, Geoffrey Irving, Georgios Piliouras. ICML 2024 (Oral).
> > 3. IP=PSPACE. Adi Shamir. J. ACM 1992.
> > 4. Probabilistic Proof Systems: A Primer. Oded Goldreich. Found. Trends Theor. Comput. Sci. 2008.
> > 5. Learning to Give Checkable Answers with Prover-Verifier Games. Cem Anil, Guodong Zhang, Yuhuai Wu, Roger Grosse. arXiv 2021.
> > 6. Computational complexity - a conceptual perspective. Oded Goldreich. Cambridge University Press 2008.
> > 7. The Complexity of Computing the Permanent. Leslie Valiant. Theoretical Computer Science 1979.
> > 8. Provably Safe Artificial General Intelligence via Interactive Proofs. Kristen Carlson. Philosophies 2021.

---

> > > ### Author Response · Authors · 2024-11-26
> > > **Follow-up on rebuttal: Reviewer W6ru**
> > >
> > > Dear Reviewer W6ru,
> > >
> > > As we approach the revision deadline (November 27th), we wanted to follow up on our rebuttal from November 18th. In our rebuttal, we addressed your thoughtful questions about novelty (clarifying the prover/verifier vs prover/disprover distinction), the role of verifiers, and probabilistic guarantees. We also committed to adding an additional experiment on Quadratic Residuosity, which goes significantly beyond GCD. We would appreciate if you could review our responses and, if you find them satisfactory, consider updating your score.
> > >
> > > Best regards,
> > > The authors

---

### Official Review · Reviewer_yycY · 2024-11-04

**Soundness:** 3
**Presentation:** 4
**Contribution:** 4
**Rating:** 8
**Confidence:** 4

**Summary:**

In this paper, the authors propose a new type of self-proving models that not just predict an output for a given input but also a proof for the correctness of the output. One of the main ideas of the paper is to use a particular notion of proof from the work of interactive proof systems in theoretical computer science, where a proof means a sequence of answers to a verifier's questions that can convince the verifier of the correctness of the output. The authors compare their approach with other similar proposals for self-proving models, and emphasise the benefit of having an instance-specific proof in their approach. They then describe two algorithms for learning such a proof-producing transformer model, namely, Transcript Learning (TL) that assumes strong supervision via successful question-answer sequences, and Reinfocement Learning from Verifier Feedback (RLVF) that does not assume such strong supervision. Their experiments with learning the GCD algorithm with a small version of GPT show the promise of their approach.

**Strengths:**

1. The idea of using a verifier from the theory of interactive proof systems for learning a self-proving model is very nice. It may lead to further interesting research activities that address the AI safety issue using several related tools from theoretical computer science, such as PCP and property testing etc.

2. The paper is written well. The discussion on related work helped me to understand what people had explored in the past, and to see the contributions of the paper more clearly. Also, the background materials are covered nicely so that I can follow most of the formal developments in the paper although I am not familiar with, for instance, interactive proof systems.

3.  The paper contains a theoretical justification, namely, Theorem 4.1. I am less confident that this theorem is useful in practice, but it is good that the authors makes an effort for proving a theoretical result. Also, their comment on the proof using the reduction to SGD and the communication complexity by a verifier (captured by the constant C) helped me to see what goes on more clearly.

**Weaknesses:**

I support the acceptance of this paper. The following points are mostly minor.

1. Having an example in addition to GCD would have convinced me of the promise of the authors' approach far more. As the authors pointed out, the proofs in this GCD case do not involve questions from the verifier, and so they are simple. Also, the annotated transcript learning is only vaguely defined, and it is only explained in terms of illustration in the example via the intermediate steps of the Euclid algorithm. Seeing one more example would have helped me to grasp what annotations would mean in other problems.

2. I suggest to include Algorithms 1 and 2 into the main text, instead of including them in the appendix. They are more or less standard, but I feel that they are one of the main contributions of the paper. Also, one unexpected thing that I found is that Algorithm 1 is derived by maximising theta over the expected probability E_{trace ~ p(trace)}[q_theta(trace)], instead of the expected log probability E_{trace ~ p(trace)}[log q_theta(trace)] (i.e., cross entropy loss). Some subtleties like this deserve the attention of the reader, I think.

**Questions:**

The only question that I have is related to what I said in the second point in the weakness box. My understanding is that Algorithm 1 uses the expected probability as a training objective, instead of expected log probability. Is there a reason for this? Is this due to the consistency with Theorem 4.1?

Here are some minor typos.

(1) L284 : EOS in Sigma^* ===> EOS in Sigma

(2) L391 : Giving examples of annotations may help some readers.

(3) L510 : Have you tried more samples in some cases and checked your conjecture?

(4) L926 : a_0 := y ===> a_0 := y^*

(5) L928 : (y,q_1^*,..., q_r^*,a_r) ===> (y^*,q_1^*,...,q_r^*)

(6) L1150 : Maybe it is better to break a line before "for s in [L_a] do"

---

> ### Author Response · Authors · 2024-11-18
>
> We sincerely appreciate your thorough and insightful review of our paper. Your comments indicate a deep understanding of our work, which makes your positive score all the more encouraging.
>
> We share your view that our paper will form a bridge through which theoretical tools could be applied to issues in AI safety. This is why we've invested considerable thought and effort towards making our writing clear and accessible to those without a background in Interactive Proof systems. We are delighted from you that hear that our efforts were successful.
>
> Below, we address your major concerns regarding: (1) the need for additional examples, (2) clarity of annotated transcript learning, (3) algorithm placement, and (4) probability optimization approach. We then address the minor corrections you identified.
>
> ### 1. Additional experiments
>
> > Having an example in addition to GCD would have convinced me of the promise of the authors' approach far more. As the authors pointed out, the proofs in this GCD case do not involve questions from the verifier, and so they are simple.
>
> We will add an additional experiment that makes use of an interactive verifier. Since this request was shared by other reviewers, we describe this experiment in the "Overall Comment." We welcome any feedback on the proposed experiment. Due to computational constraints, we may not by able to conclude the experiments by the end of the discussion period, but we will include the results in the camera ready version of the paper.
>
> ### 2. Clarity of Annotated Transcript Learning
>
> > Also, the annotated transcript learning is only vaguely defined, and it is only explained in terms of illustration in the example via the intermediate steps of the Euclid algorithm. Seeing one more example would have helped me to grasp what annotations would mean in other problems.
>
> We address this concern in two ways:
>
>  - We've added a new paragraph to Section 4.3 (immediately before Section 5) that explains annotations through an analogy to Chain-of-Thought reasoning. This provides readers with a familiar framework for understanding our approach.
>  - The new Quadratic Residuosity experiment will demonstrate annotations in a completely different context from GCD. The annotations here involve tracking mathematical properties of group elements different from the Euclidean algorithm steps.
>
> ### 3. Algorithm placement
>
> > I suggest to include Algorithms 1 and 2 into the main text, instead of including them in the appendix. They are more or less standard, but I feel that they are one of the main contributions of the paper
>
> We agree these algorithms are central contributions. Our solution aims to balance accessibility with space constraints:
>
> - For the conference version: We've added hyperlinks in the main text (particularly around Theorem 4.1) that let readers quickly access the algorithm specifications.
> - For the full version (preprint and journal): We will integrate both algorithms into the main text with expanded discussion.
>
> ### 4. Optimization objective in Algorithm 1
>
> > Also, one unexpected thing that I found is that Algorithm 1 is derived by maximising theta over the expected probability [...], instead of the expected log probability [...] Some subtleties like this deserve the attention of the reader, I think [...] Is there a reason for this? Is this due to the consistency with Theorem 4.1?
>
>
> Excellent observation. The key to answering this question is by drawing a distinction between the *objective* and the *implementation* of the algorithm:
>
>  - The Objective: Optimizing the Verifiability of the learned model through what we call the "agreement function" (based on expected probability).
>  - The Implementation: We'd want to optimize the agreement function directly, but how do we compute the gradients? Lemma B.4 is the answer: we can achieve this indirectly by accumulating gradients from the cross-entropy loss computed at each token - similar to how language models are typically trained.
>  - The Connection: Algorithm 1's objective is to optimize the agreement (expected probability), but does so by taking gradients through the log probability and accumulating them. This matches your intuition about cross-entropy loss!
>
> Thank you for helping us reach this clarification; we have added it to the paper immediately after Theorem 4.1. Does this answer your question and address your concern?
>
> ### 5. Response to minor points
>
> Thank you for your careful attention to detail.
>
> 1. L284: Corrected.
> 2. L391: Added clarifying examples of annotations (see major point 2).
> 3. L510: Once our GPU is free from the new Quadratic Residue experiment, we will repeat the Base of Representation experiments on additional samples. We will include the results in the camera-ready version of the paper.
> 4. L926: Corrected.
> 5. L928: Corrected.
> 6. L1150: Added line break.

---

> > ### Comment · Reviewer_yycY · 2024-11-23
> > **Thanks!**
> >
> > Thank you for your detailed explanation. It improved my understanding of the paper. Also, it is good that you commit to apply your approach to another more involved example.

---

### Author Response · Authors · 2024-11-18
**Overall comment: Additional experiment**

Dear all,

In response to reviewers TyWJ, W6ru and yycY, we intend to run an additional experiment going beyond the current GCD setup. With GCD, the verification algorithm was deterministic and non-interactive—allowing experiments to run on a single GPU within days (excluding RLVF).

We propose to extend our work with experiments on the Modular Square Root (MSqrt) problem. MSqrt is considered substantially more challenging than GCD, and is a key problem in various classical cryptosystems. We detail our proposed experiment below and welcome your feedback.

Thanks to our flexible codebase, we can implement these new experiments without restructuring our Self-Proving GPT. However, given the increased computational complexity of Quadratic Residue, we'll need to scale up both model size and training data/iterations. While the runs of these experiments will extend beyond the discussion period, we commit to including them in the camera-ready version.

Thank you,
The authors

### The Modular Square Root (MSqrt) problem
We say that an integer $x$ has a quadratic residue (mod another integer $N$) if there exists $y$ such that $x = y^2 \mod N$. In the Modular Square Root (MSqrt) problem, the input is a pair of integers $(x, N)$ and the output is one of the following:
1. Either any $y$ such that $x = y^2 \mod N$, if such $y$ exists.
2. A special symbol $\bot$ if no such $y$ exists (i.e., if $x$ is not a quadratic residue).

Technical comment: For any $x \notin \{0,1\}$, if $x = y^2 \mod N$ then it is also the case that $x = (-y)^2 \mod N$. Thus, MSqrt is not a function with unique correct outputs but rather a relation where inputs may have multiple correct outputs. We address this generalization in our newly-added Appendix A, which reveals interesting connections to loss functions and proof systems literature. We're happy to elaborate on this generalized definition if needed.

### An Interactive Proof system for MSqrt
Intuition: For a given $(x,N)$, if $y \in MSqrt(x,N)$ where $y \neq \bot$, then the prover simply sends $y$ to the verifier, who can verify correctness by checking that $y^2$ has the same residue mod $N$ as $x$ (this is an easy computation when $N$ is prime and gets increasingly hard with composite $N$). The more interesting case occurs when the prover claims $x$ has no quadratic residue ($y = \bot$), which triggers the interactive protocol for Quadratic Nonresiduosity (Goldwasser, Micali and Rackoff, 1985).

We present the verifier for this protocol next (completeness and soundness proofs available upon request):

**Verifier $V$: Takes input (x, N) and interacts with a Prover $P$**
1. Receive an (allegedly correct) output $y \in \mathbb{N} \cup \{\bot\}$ from the Prover.
2. If $y \neq \bot$: Accept if and only if $x = y^2 \mod N$.
3. Else ($y = \bot$): Sample a random $r \in \{1, \dots, N-1}$ and a bit $b \in \{0, 1\}$.
4. Send $q = x^b \cdot r^2 \mod N$ to the prover.
5. Receive a response $a$ from the prover.
6. If $b = 1$: Accept if and only if $a = y = \bot$.
7. Else ($b = 0$): Accept if and only if $q = a^2 \mod N$.

Technical comment: we note that Quadratic Residuosity as we defined it above admits a noninteractive (determinstic) proof system. However, we chose this proof system to explore Self-Proving models in an interactive setting, as pondered by the reviewers. As an aside, we remark that in this proof system manages, the verifier is convinced (with high probability) of $x$ being a non-square *without revealing the factorization of $N$*, i.e., it is a zero knowledge proof of $x$ being a quadratic non-residue.

### Experiments on MSqrt
We will repeat the TL and RLVF experiments on MSqrt. As with the GCD, we expect non-annotated TL to have moderate efficacy, which can then either be boosted with RLVF or with annotations. Informed by our Annotated TL experiments on GCD, we will add annotations derived from intermediate computations in the honest prover's strategy (e.g., a factorization of $N$ can be used by the prover to efficiently compute MSqrt).

---

### Author Response · Authors · 2024-11-22
**Rebuttal follow-up**

Dear reviewers,

This is a gentle reminder of our rebuttals, which we posted on November 18th. If you have any remaining concerns or questions, please let us know. If your concerns have been addressed, we humbly ask you to consider updating your scores.

We look forward to continuing the discussion of our paper.

Best regards,
The authors

---

### Meta-Review · Area_Chair_d74S · 2024-12-25

**Metareview:**

The paper introduces a framework when ML models can prove the correctness of their output. The notion of proof here is "interactive proofs" as studied in computational complexity, with a polynomially-bounded verifier and a potentially unbounded prover that can interact. When the answer is correct, then the prover should be able to convince the verified with high probability of its correctness, on the other hand when the answer is wrong the verifier should not accept the answer with high probability. The contribution of this paper is not related to interactive proof systems, they are using the standard notion. The contribution is mainly to introduce this framework to modern ML settings. This requires the training process to be aware of a specific verifier that will be used and the training process needs to be augmented with transcripts. The authors propose a gradient-descent based approach as well as an RLVF approach. The paper definitely introduces new ideas, however, the assumptions required for learning algorithms to converge are fairly strong -- this is to be expected-- and also unrealistic. Thus, there needs to be a thorough empirical evaluation. The authors have presented experiments with GCD in the main paper, and subsequently performed experiments with discrete square root during the rebuttal period.

**Additional Comments On Reviewer Discussion:**

Two of the reviewers interacted well with the authors; these were also the most informative reviews and I had written my meta review using those reviews and the paper itself.

---

### Decision · Program_Chairs · 2025-01-22

Reject